# Squeezing SGD Parallelization Performance in Distributed Training Using Delayed Averaging

## Abstract

The state-of-the-art deep learning algorithms rely on distributed training systems to tackle the increasing sizes of models and training data sets. Minibatch stochastic gradient descent (SGD) algorithm requires workers to halt forward/back propagations, to wait for gradients aggregated from all workers, and to receive weight updates before the next batch of tasks. This synchronous execution model exposes the overheads of gradient/weight communication among a large number of workers in a distributed training system. We propose a new SGD algorithm, DaSGD (Local SGD with Delayed Averaging), which parallelizes SGD and forward/back propagations to hide $100\%$ of the communication overhead. By adjusting the gradient update scheme, this algorithm uses hardware resources more efficiently and reduces the reliance on the low-latency and high-throughput inter-connects. The theoretical analysis and the experimental results show its convergence rate $O(1/\sqrt{K})$, the same as SGD. The performance evaluation demonstrates it enables a linear performance scale-up with the cluster size.

## 1 Introduction

Training deep learning models using data parallelism on a large-scale distributed cluster has become an effective method for deep learning model training. The enormous training data set allows a huge batch of training tasks on different data samples running in parallel. The pinnacle of this method reduces the training time of the benchmark ResNet-50 from days to a couple of minutes (Goyal et al., 2017; You et al., 2017b; Akiba et al., 2017; You et al., 2017a; Ying et al., 2018; Goodfellow et al., 2016). During Mini-batch stochastic gradient descent (SGD), these workers have to halt, wait for the computed gradients aggregated from all of the workers and receive a weight update before starting the next batch. The wait time tends to worsen when the number of workers increases. As the workloads are spread over a larger cluster, the computation time is greatly shortened and the communication overheads take a larger portion of the overall cost.

System designers address this concern by improving inter-chip connects with higher throughput and lower latency and refining network topology (Li et al., 2019), such as NVIDIA DGX-1 (NVI, 2017) and NVIDIA DGX-2 (NVS, 2018). Additional care has been given to reduce the intermediate steps that would increase communication latency. These methods effectively reduce the wait time during Mini-batch SGD on a large-scale distributed system (Gau, 2019).

Communication efficient SGD algorithms (Lin et al., 2017; Wangni et al., 2018; Alistarh et al., 2018; Dean et al., 2012; Recht et al., 2011; Zhang et al., 2015; Wang & Joshi, 2018; Wang & Joshi, 2018; Lin et al., 2018) are proposed to reduce the communication requirements. A successfully modified SGD algorithm shows their convergence rates comparable to Mini-batch SGD through both theoretical analysis and experimental results. Another challenge is to demonstrate good performance evaluation results based on the common large distributed training system setups.

A modern data center design prefers cost-efficient hardware blocks and a balanced configuration for the typical workloads (Barroso et al., 2018). Under these workloads, hardware resources are utilized in a balanced fashion. A distributed training system works in the opposite manner. During forward/back propagations, the computing resources are fully used while the system inter-connects

are completely idle. During SGD, the computing resources are mostly idle while the system inter-connects are throttled at the peak throughputs. The performance of distributed training systems may be improved in addition to better hardware. That is, training workloads may be re-structured for balanced utilization of the hardware resources.

Inspired by the modern system design practices, we propose DaSGD, a new SGD method, enabling SGD running parallelly with forward/back propagation and balanced utilization of the hardware resources. It replaces a Mini-batch SGD with Local SGD iterations. In a Local SGD (Lin et al., 2018; Wang & Joshi, 2018), each worker evolves a local model by performing sequential SGD updates, before synchronizing by averaging models among the workers. DaSGD uses Local SGD to add weight synchronization points and allows weights to be updated between Local SGD iterations. Based on the network throughput and the model size, it schedules delayed model averaging for a defined number of Local SGD iterations, which allows workers to compute the next batch while the weights are transferred over a large distributed cluster. The theoretical analysis shows its convergence rate is $O(1/\sqrt{K})$, where $K$ is the iteration step, same as Mini-batch SGD.

The main contributions of this paper are the following.

- We present DaSGD as an algorithm-system co-design method for a large-scale distributed training system. It enables designing a more balanced and better-utilized system, more than a new variant of the SGD algorithm. The discussions and analyses in this paper are organized around its equivalency to the traditional SGD and its benefit from the system design perspective.

- We provide the theoretical analysis of its convergence rate. It shows the convergence rate at $O(1/\sqrt{K})$, the same as Mini-batch SGD.

- Our experiments focus on the training progresses (in terms of loss and accuracy) at the epoch level. It shows DaSGD allows the training converges at the same rate of SGD epoch by epoch, which is a good indicator of statistical efficiency and the eventual time-to-convergence. The experiments also explore the proper ranges of these parameters.

- A performance evaluation of real-life systems measures many performance issues in a system, such as the reduction algorithm, GPU interconnect topology and throughputs, network throughputs, which are out of the scope of our discussion. Instead, we abstract an analytical model from the key performance parameters of the system configuration and the training setup. The analytical model demonstrates DaSGD introduces a linear performance scale-up with the cluster size.

## 2 BACKGROUND

### 2.1 STOCHASTIC GRADIENT DESCENT

Stochastic Gradient Descent (SGD) is the backbone of numerous deep learning algorithms (Ghadimi & Lan, 2013). Supervised deep learning demands massive training datasets. Training a deep learning model needs many epochs for training to converge. A variant of classic SGD, synchronous mini-batch SGD (Bottou, 2010), has become the mainstream due to a faster convergence rate., supported by prevalent machine learning frameworks, such as Tensorflow (Abadi et al., 2016), Pytorch (Paszke et al., 2019), MxNet (Chen et al., 2015).

Mini-batch SGD as a weight update process is shown in Eq. 1.

$$x_{k+1} = x_k - \frac{\eta}{B} \sum_{j=1}^{B} \nabla F(x_k, s_k^{(j)}) \tag{1}$$

where $x \in \mathbb{R}^d$ is the weight of a model, $\eta$ is the learning rate, $B$ is the batch size, $\mathcal{S}$ is the training dataset, $s_k^{(j)} \subset \mathcal{S}$ is a random sample, $\nabla F(x_k, s_k^{(j)})$ is the stochastic gradient given the sample $s_k^{(j)}$.

From a system perspective, a distributed training system may compute a batch of gradients on several workers. At the end of a batch, a reduction operation is performed on the gradients on a worker first and a worker sends out only a copy of local averaged gradients. Further reductions are performed on gradients from different workers until a final copy of the averaging gradients is obtained. The above

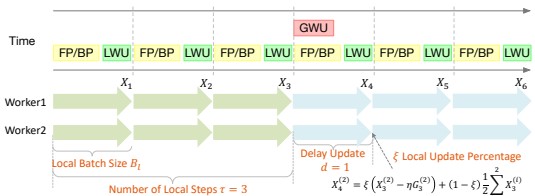

Figure 1: A timing diagram of DaSGD.

equation may be rewritten as

$$x_{k+1} = x_k - \frac{\eta}{M} \sum_{j=1}^{B} g(x_k, s_k^{(j)})$$

(2)

where $M$ is the number of workers, $g(x_k, s_k^{(j)})$ is the stochastic gradient that worker $j$ aggregates locally for that batch.

$$g(x_k, s_k^{(j)}) = \frac{M}{B} \sum_{i=1}^{\frac{B}{M}} \nabla(x_k, s_k^{(i)})$$

(3)

## 2.2 COMMUNICATION EFFICIENT SGD ALGORITHMS

### 2.2.1 ASYNCHRONOUS SGD

There are a few asynchronous training methods, such as Downpour SGD (Dean et al., 2012), Hogwild (Recht et al., 2011), Elastic Averaging SGD (Zhang et al., 2015). In these models, every worker has its own copy of weights. A worker performs forward propagation and back propagation on its partition of samples, and then sends the calculated gradients asynchronously to a pool of parameter servers that manage a central copy of weights. The parameter servers update the central copy and then send the new weights asynchronously to each worker. While each worker communicates gradients at a different time and avoids congestions at worker inter-connects, the parameter servers might be a performance bottleneck. For non-convex problems, ASGD requires that the staleness of gradients is bounded (Lian et al., 2015) to match the convergence rate $O(1/\sqrt{K})$ of synchronous SGD, where $K$ denotes the total Iteration steps.

### 2.2.2 LOCAL SGD

Another set of methods targets at reducing the frequency of inter-worker communication and is called periodic averaging or Local SGD (Wang & Joshi, 2018; Wang & Joshi, 2018; Lin et al., 2018). A worker performs SGD on its local copy of weights for $\tau$ times, where $\tau$ denotes the local iteration steps. After $\tau$ local updates, local copies are synchronously averaged across all workers globally. Several works suggested that Local SGD incurs the same convergence rate $O(1/\sqrt{K})$ as SGD. The total number of steps to train a model remains similar but the total amount of inter-worker communication is reduced by $\tau$ times. This has a similar effect as training with a large batch size, where the number of synchronizations decreases with an increase of batch size. With Local SGD, SGD and forward/back propagations are still blocking while system resources are unbalanced.

## 3 DASGD

In this paper, we propose a new algorithm, called *Local SGD with Delayed Averaging*, *DaSGD* for short. It aggregates gradients and updates weights in a relaxed manner, which helps parallelize the computation of forward/backward propagation with two other execution components: the execution of global weight averaging and inter-worker data communication.

Our algorithm was initially inspired by the Local SGD algorithm (discussed in Section 2). Although Local SGD was designed to reduce communication and synchronization overhead, it still involves a significant amount of communication overhead. To further decrease communication overhead, even to zero, the proposed algorithm exploits a delayed averaging approach that makes two novel improvements based on Local SGD. First, in order to merge remote weights by other workers with local in a deterministic way, DaSGD serializes forward propagations and back propagations for different samples. Second, workers start with local computations for the next local batch while waiting for the aggregation and synchronization of global weights. In this way, the global communication

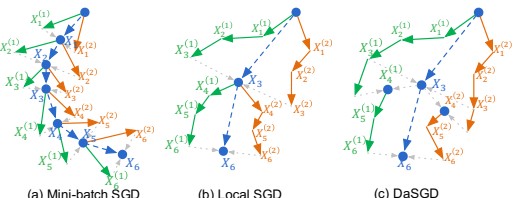

Figure 2: Loss landscape of (a) Mini-batch SGD, (b) Local SGD, and (c) DaSGD. 12 samples are updated on two workers. The orange and green arrows represent the updated loss function of each sample, and the blue arrows describe the location of the updated loss function on the global model.

and synchronization overhead are hidden or overlapped by local computations at the cost of a delayed update of local weights. However, theoretically we will prove that the convergence rate is the same as Mini-batch SGD. Furthermore, DaSGD parameterizes the overlapping degree so that when a large training cluster requires a longer time to synchronize, a worker may perform more iterations of local computations.

Fig. 1 illustrates the proposed algorithm by showing a wall-clock time diagram of 6 training iterations. There are 2 workers, dividing a batch into 2 local batches. Each worker computes 6 local batches. Each local batch contains $B_l$ samples. Each worker maintains a local copy of model. According to Local SGD, for a local batch, each worker operates $B_l$ forward/backward propagations and then updates the weights of its local model. After 3 local updates, a worker synchronizes local weights with the other workers, resulting in an AllReduce operation being generated to average the model weights. In Fig. 1, all workers wait for, at local iteration 3, the global synchronization to be finished and then start to operate on the next local batch each, in the scenario of Local SGD.

DaSGD implements a key feature by imposing delay update on Local SGD. As shown in Fig. 1, a worker, at local step 3, broadcasts its local weights to the wild and then immediately starts to compute on the next local batch, without waiting for the global synchronization to be finished. Later, at local iteration 4, the worker receives all the other workers' weights and then updates its local weights. This design very efficiently overlaps the communication of weights and forward/backward propagations of the next local batch.

In DaSGD, we use $\tau$ to denote the number of local batches between two consecutive global synchronizations. Therefore, $\tau$ is a controlling parameter that quantifies the number of propagations between weight averaging globally. During the delay update, both local computation and the global communication of weights are executed in parallel. As long as communication time is no more than the computation time of $d$ local iterations, the communication time can be hidden in the overall model training time. Careful tuning of $d$ and $\tau$ can realize full parallelism of global averaging and local computations. Unlike Local SGD, $\tau$ does not have to be large, as it is not only used to reduce inter-worker communication overhead (Lin et al., 2017).

In the following part of this section, in order to compare the proposed algorithm and traditional SGDs, we start with the update framework of each algorithm, and then qualitatively analyze execution time. Finally, we discussed the updated rules and the convergence rate in detail.

### 3.1 UPDATE FLOW OF DIFFERENT SGD

Fig. 2 explains the mechanisms of weight update flows of *Mini-batch SGD*, *Local SGD*, and DaSGD by taking an example of a 2-worker parallel training process that sets the batch size as 2 samples. The 2 workers are distinguished by yellow and green arrows. In the Mini-batch SGD (as shown in Fig. 2(a)), every worker updates its local weights once every *mini-batch*, which is computed as the batch size divided by the number of workers. When both workers finish local updates for a mini-batch, local weights are merged to compute their average (shown by blue arrows). Next, both workers update their local weights with the average. Local SGD (shown in Fig. 2(b)) reduces the weight aggregation times by letting every worker first update weights locally for continuous $\tau$ *local batch*es in a row before a global merge is made. Local batch in the context of Local SGD is just a synonym of the mini-batch in the context of Mini-batch SGD.

Same as the regular periodic averaging method (i.e., Local SGD), in the proposed algorithm, each worker updates local weights for $\tau$ local batches before a global aggregation. A novel change made by the proposed algorithm is to delay weight update from global to local after the global averaging. A worker may delay the update for $d$ *step*s (i.e., samples) of local weight updates ($d = 1$ in

this example, as shown in Fig. 2(c)). With this novel algorithmic design, the time of global weight averaging can be hidden by parallelizing it with local computation by a worker, i.e., forward propagation, backward propagation, and local weight update. Large $d$ can be set if the time of global weight aggregation is very long in a large-scale distributed training to shorten overall training time.

## 3.2 CONVERGENCE ANALYSIS

In this section, we provide a theoretical analysis of the DaSGD algorithm. We will prove the convergence rate for DaSGD on non-convex problems and show that it converges at the same rate $O(1/\sqrt{K})$ as Mini-batch SGD and Local SGD. To facilitate the convergence analysis, we firstly introduce the assumptions.

### 3.2.1 ASSUMPTIONS

The convergence analysis is under the assumptions as the following, which are similar to the Local SGD (Wang & Joshi, 2018):
- Lipschitzian gradient: $|| \bigtriangledown F(x) - \bigtriangledown F(y)|| \leq L||x - y||$
- Unbiased gradients: $E_{\mathcal{S}_k|x}[g(x)] = \bigtriangledown F(x)$
- Lower bounder: $F(x) \geq F_{inf}$
- Bounded variance: $E_{\mathcal{S}_k|x}||g(x) - \bigtriangledown F(x)||^2 \leq \sigma^2$
- Independence: All random variables are independent to each other
- Bounded age: The delay is bounded, $d < \tau$

where $\mathcal{S}$ is the training dataset, $\mathcal{S}_k$ is set $\left\{s_k^{(1)}, ..., s_k^{(M)}\right\}$ of randomly sampled local batches, $L$ is the Lipschitz constant.

### 3.2.2 UPDATE RULE

$$x_{k+1}^{(m)} = \begin{cases} \xi x_k^{(m)} - \eta\xi g\left(x_k^{(m)}\right) + \frac{(1-\xi)\sum\limits_{j=1}^{M}\left[x_{k-d}^{(j)} - \eta g\left(x_{k-d}^{(j)}\right)\right]}{M} \\ \quad , \quad (k+1) \mod \tau = d \\ x_k^{(m)} - \eta g\left(x_k^{(m)}\right), \quad \text{otherwise} \end{cases} \tag{4}$$

where $x_k^{(m)}$ is the weights of worker $m$ at $k$-th iteration, $\eta$ the learning rate, $M$ the number of workers, and $g(x_k^{(m)})$ the stochastic gradient of worker $m$. For every $k$ that satisfies $(k+1) \mod \tau = d$, a global average is updated to local weights. Besides, $\xi$ is an auxiliary parameter to adjust the weight of local weights in contrast to the global average when fusing them together.

From equation 4, we can define the average weight and the average gradient

$$\mu_k = \frac{1}{M}\sum_{i=1}^{M} x_k^{(i)}, \ \bar{g}_k = \frac{1}{M}\sum_{i=1}^{M} g\left(x_k^{(i)}\right).$$

Reformatting it, the update rule for the average weight is

$$\mu_{\tau(k+1)+d} = \mu_{\tau k+d} - \eta\left[\xi\sum_{i=\tau-d}^{\tau-1}\bar{g}_{\tau k+d+i} + \sum_{i=0}^{\tau-1-d}\bar{g}_{\tau k+d+i}\right]$$

It is observed that the averaged weight $\mu_{\tau(k+1)+d}$ is performing a perturbed stochastic gradient descent. Thus, we will focus on the convergence of the averaged weight $\mu_{\tau(k+1)+d}$, which is a common approach in the literature of distributed optimization (Wang & Joshi, 2018; Wang & Joshi, 2018). SGD can converge to a local minimum or saddle point due to the non-convex objective function $F(x)$. Therefore, the expected gradient norm is used as an index of convergence.

### 3.2.3 CONVERGENCE RATE

The $\epsilon$-suboptimal solution of the algorithm is $\mathbb{E}_k\left[\frac{1}{K}\sum_{k=1}^{K}\|\bigtriangledown F(\mu_k)\|^2\right] \leq \epsilon$. The learning rate is usually set as a constant and is decayed only when the training process is saturated. Therefore, we analyze the case of a fixed learning rate and study the lower limit of error at convergence.

**Theorem (Convergence of DaSGD).** Under assumptions, if the learning rate satisfies $2L\eta d\xi^2 - \xi + \frac{6\xi L^2\eta^2 d + 6L^2\eta^2(\tau-d)}{1-\xi^2} + 6\xi L^2\eta^2 d \leq 0$ and $2L\eta(\tau-d) - \xi + \frac{6\xi L^2\eta^2 d + 6L^2\eta^2(\tau-d)}{1-\xi^2} + 6\xi L^2\eta^2 d + 6L^2\eta^2(\tau-d) \leq 0$ at the same time, the average-squared gradient norm after $K$ iterations is bounded as follows

$$\mathbb{E}\left[\frac{1}{K}\sum_{k=1}^{K}\|\triangledown F(\mu_k)\|^2\right] \leq \frac{2\left[F(\mu_1)-F_{inf}\right]}{\eta K(\xi d+\tau-d)} + \frac{\eta}{M}\frac{2L\sigma^2\left[\xi^2 d+\tau-d\right]}{(\xi d+\tau-d)} + \eta^2\frac{6L^2(1+\xi)}{\xi d+\tau-d}\sum_{l=\tau-d}^{\tau-1}\sum_{i=0}^{l}\sigma^2 + \eta^2\frac{6d\xi^2L^2\tau\sigma^2(1+\xi)}{(\xi d+\tau-d)(1-\xi^2)}$$
$$+\frac{\eta^2}{K}\frac{12d\sigma^2 L^2\xi^2(\tau-d)(1+\xi)}{(\xi d+\tau-d)(1-\xi^2)} + \frac{\eta^2}{KM}\frac{12L^2 d\xi^2(\tau-d)(1+\xi)}{(\xi d+\tau-d)(1-\xi^2)}\sum_{i=1}^{d-1}\|\triangledown F(\mathbf{X}_i)\|_F^2$$

where $\boldsymbol{X}_k = \left[x_k^1,...,x_k^m\right]$, $\|\ \|_F^2$ is the Frobenius norm. All proofs are provided in the Appendix.

**Corollary.** Under assumptions, if the learning rate is $\eta = \frac{M+V}{M}\sqrt{\frac{M}{K}}$ the average-squared gradient norm after $K$ iterations is bounded by

$$\mathbb{E}\left[\frac{1}{K}\sum_{k=1}^{K}\|\triangledown F(\mu_k)\|^2\right] \leq \frac{2\left[F(\mu_1)-F_{inf}\right]+2L\sigma^2\left[\xi^2 d+\tau-d\right]}{\sqrt{MK}(\xi d+\tau-d)} + \frac{M^2}{K^3}\left(1+\frac{V}{M}\right)^4\frac{6L^2(1+\xi)}{\xi d+\tau-d}\sum_{l=\tau-d}^{\tau-1}\sum_{i=0}^{l}\sigma^2$$
$$+\frac{M^2}{K^3}\left(1+\frac{V}{M}\right)^4\frac{6d\xi^2 L^2\tau\sigma^2(1+\xi)}{(\xi d+\tau-d)(1-\xi^2)} + \frac{M^2}{K^3}\left(1+\frac{V}{M}\right)^4\frac{12d\sigma^2 L^2\xi^2(\tau-d)(1+\xi)}{(\xi d+\tau-d)(1-\xi^2)}$$
$$+\frac{M}{K^3}\left(1+\frac{V}{M}\right)^4\frac{12L^2 d\xi^2(\tau-d)(1+\xi)}{(\xi d+\tau-d)(1-\xi^2)}\sum_{i=1}^{d-1}\|\triangledown F(\mathbf{X}_i)\|_F^2$$

If the total iterations $K$ is sufficiently large, then the average-squared gradient norm will be bounded by

$$\mathbb{E}\left[\frac{1}{K}\sum_{k=1}^{K}\|\triangledown F(\mu_k)\|^2\right] \leq \frac{2\left[F(\mu_1)-F_{inf}\right]+2L\sigma^2\left[\xi^2 d+\tau-d\right]}{\sqrt{MK}(\xi d+\tau-d)}$$

Therefore, on non-convex objectives, for the total iterations $K$ large enough, DaSGD converges at rate $O(1/\sqrt{K})$ consistent with the Mini-batch SGD and the Local SGD. Besides, it is worth noting that these three introduced auxiliary parameters ($\tau$, $d$ and $\xi$) are mostly in the high-order term $O(1/K^2)$ and $O(1/K^3)$, which has little influence on the convergence of DaSGD.

### 3.3 Guidelines for Using DaSGD

DaSGD is identical to Local SGD except that the global model is updated for delayed $d$ local steps. The tuning of the other parameters is the same as that of Local SGD. Therefore, in this paper, we merely discuss how to set this key delay parameter.

In order to overlap communication with computation in DaSGD, the weight/gradient transfer time $t_c$ across workers is required to be shorter than the time of $d$ local iterations, that is, $t_c < d \times t_p$, where $t_p$ is the computation time of one iteration. For deep learning systems, the weight transfer time $t_c$ across multiple workers is approximately calculated as $t_c = m \times n_p/\text{BW}$, where $n_p$ denotes the number of parameters of the model, $m$ the number of workers, and BW the bandwidth of the device. The computation time $t_p$ of one local batch is approximately calculated as $t_p = B_l \times \text{FLOP}/\text{FLOPS}$, where FLOP denotes the floating-point operation counts of local computation, $B_l$ the local batch size, FLOPS denotes the computation speed of the device in terms of floating-point operations per second (FLOPS).

In a nutshell, the delay is given by $d > \frac{t_c}{t_p} = \frac{m \times n_p \times \text{FLOPS}}{B_l \times \text{BW} \times \text{FLOP}}$. Note that the delay is highly related to the structure of neural network models, e.g., the number of parameters and FLOP, and the configurations of deep learning training (local batch, worker number, bandwidth of the device and computation speed). With the rapid improvement of the bandwidth and performance of distributed training, the mode size of the neural network grows relatively slowly. As a result, in most cases, when the delay is one single iteration, i.e. $d = 1$, the weight transfer can be processed completely in parallel with local updates. In addition, as the worker number increases, the transfer time increases, and hence the delay would increase moderately accordingly. The cooperative design of various parameters in DaSGD and hardware is discussed in detail in the following sessions.

Table 1: Validation Accuracy of DaSGD, Mini-batch SGD and Local SGD on CIFAR-10.

| Model | Validation accuracy after 50 epochs | | | |
| --- | --- | --- | --- | --- |
| | Mini-batch SGD | | 2*Local SGD | 2*DaSDG |
| | 32 batch | 1024 batch | | |
| GoogleNet (Szegedy et al., 2015) | 0.9467 | 0.9409 | 0.9468 | 0.9444 |
| VGG-16 (Simonyan & Zisserman, 2014) | 0.9362 | 0.9264 | 0.9330 | 0.9343 |
| ResNet-50 (He et al., 2016) | 0.9113 | 0.9037 | 0.9062 | 0.9088 |
| ResNet-101 | 0.9116 | 0.9019 | 0.9061 | 0.9045 |
| DenseNet-121 (Huang et al., 2017) | 0.9367 | 0.9332 | 0.9369 | 0.9357 |
| MobileNetV2 (Sandler et al., 2018) | 0.9230 | 0.9304 | 0.9241 | 0.9304 |
| ResNeXt29 (Xie et al., 2017) | 0.9475 | 0.9403 | 0.9424 | 0.9415 |
| DPN-92 (Chen et al., 2017) | 0.9507 | 0.9354 | 0.9513 | 0.9502 |

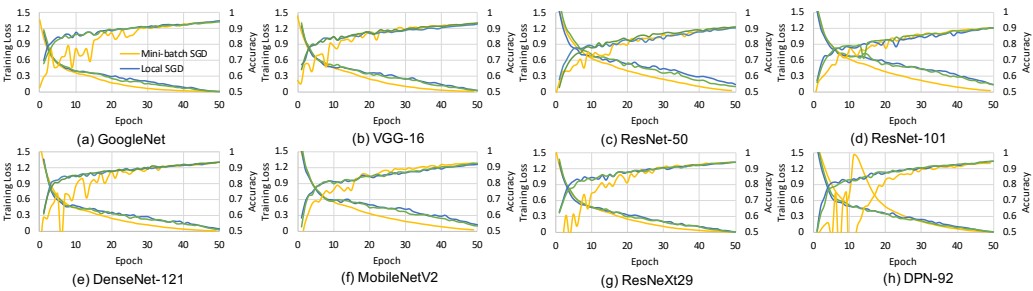

Figure 3: Validation loss(dotted line), training loss, and validation accuracy(solid line) of different models on the CIFAR-10 dataset.

## 4 EXPERIMENTAL RESULTS

In this section, we will first introduce experimental settings, then the comparison of convergence for different models, and last the effect of different parameters for DaSGD convergence.

### 4.1 EXPERIMENTAL SETUP

We implemented our system based on the Fast.Ai (Howard, 2018) platform upon CIFAR-10 dataset. The learning rate is linearly increased from $0.0001$ to $0.01$ in the first $30\%$ epochs and then decreased from $0.01$ to $0.0001$ in latter $70\%$ epochs, by respecting *One Cycle Policy* (Smith, 2017). A larger learning rate prevents the model from falling in the steep area of the loss function to find a flat minimum, while a smaller learning rate prevents training from diverging and converging to a local minimum. The proposed learning rate method assists in realizing high validation accuracy quickly (in a few iterations) so that we could only set total training epochs as 50 for a comparison between DaSGD and two other alternatives, Local SGD and Mini-batch SGD. The weight decay (commonly called L2 regularization) is $0.01$. And the momentum of SGD, which accelerates gradients vectors drops in the right direction, is $0.9$.

### 4.2 CONVERGENCE RATE AND ACCURACY COMPARISON

We compare DaSGD with Mini-batch SGD and Local SGD in terms of convergence rate and accuracy upon a set of neural network models that are trained for 50 epochs for CIFAR-10 dataset. All models are trained with 32 workers. For the three alternatives, the batch size is 1024 and the total number of iterations is 2450. For Mini-batch SGD, according to data parallelism, the mini-batch size of each worker is 32. For Local SGD and DaSGD, the local batch size $B_l$ is 32, which equals to the mini-batch size of Mini-batch SGD [1]. For Local SGD and DaSGD, the number of local steps $\tau$ is 4. For DaSGD, the delayed update steps $d$ is 1.

TABLE 1 first demonstrates that high validation accuracy up to $0.9513$ can be obtained with only $2450$ iterations with $1024$ of batch size, even without tuning hyper-parameters. For Mini-batch SGD, due to large batch size, the hyper-parameters need to be tuned carefully. Large batch size training

---

[1]The mini-batch size is a concept in Mini-batch SGD; the local batch size is a concept of Local SGD. They equal to each other and are computed as the batch size divided by the number of workers.

leads to great accuracy loss. The hyper-parameter recipe for large-batch training is complex and tedious, and the algorithms based on local update (Local SGD and DaSGD) overcome this problem, since the batch size in local updating is small $(B/m)$, which is $B$ in Mini-batch SGD. Thus, without any hyper-parameter adjustment for large-batch training, in addition to MobileNetV2, the validation accuracy of Local SGD and DaSGD is higher than that of the Mini-batch SGD. Fig. 3 shows this more clearly. At the beginning of the training, since the batch size is large, the algorithm based on Mini-batch SGD is usually very unstable, and the validation accuracy fluctuates greatly. And, the convergence rate of Mini-batch SGD is slower than that of the Local SGD and DaSGD. At the end of the training, although the training loss of Mini-batch SGD is smaller, Local SGD and DaSGD have small test loss and higher validation accuracy. In addition, compared with Local SGD, the validation accuracy of DaSGD is slightly reduced due to delayed averaging.

## 4.3 PARAMETER EFFECTS OF DASGD

We evaluate the effects of 5 parameters with respect to the convergence rate and validation accuracy for the ResNet-50 model based on DaSGD in Fig. 4. 5 parameters are the number of workers $m$, the local batch size $B_l$, the number of local steps $\tau$, the local update percentage $\xi$ and the delay factor $d$. We study every parameter by only tuning one at a time, with the rest set as the baseline setting: $m = 32$, $\tau = 4$, $d = 2$, $B_l = 32$, and $\xi = 0.25$.

**Worker number** $m$. Fig. 4(a) suggests that DaSGD has a fast convergence rate and high validation accuracy in general for any number of workers tested. As the number of workers increases from 2 to 256, the convergence rate becomes lower and validation accuracy degrades. Since the local batch size remains unchanged as 32, when the worker number is 256, the total batch size has reached 8192 ($32 \times 256$), which results in a decrease of validation accuracy of around $2\%$ and an increase of training loss of 0.31. Since the number of local steps is 4, DaSGD communicates once for a total number of samples of $32K$ ($32 \times 256 \times 4$). The CIFAR-10 dataset has only 50000 training samples, so the number of communications is just 2 for this large number of workers.

**Local batch size** $B_l$. Fig. 4(b) illustrates that DaSGD exhibits a good convergence rate with moderate local batch sizes. When the local batch size is large as 256 or small as 8, the validation accuracy is significantly reduced. Interestingly, the same observation is obtained in Mini-batch SGD. The phenomena are attributed to that large local batch size, though reducing the overall iterations, leads to poor generation ability, while small-batch size reduces the generalization error due to noise, but requires a large number of iterations. Hence, the selection of the right local batch size is critical to the performance of DaSGD. Based on our tests in Fig. 4(b), 32 or 64 of local batch size is the setting of choice, resulting in the best validation accuracy and training loss when comparing to the rest of values.

It is worth noting that the total batch size is computed as $B = m \times B_l$. When the worker number is 32 and the local batch size is 256, the total batch size rises to 8k, which faces the same challenge of tuning hyper-parameters of large-batch training discussed above. To obtain a good convergence rate for such a large batch size needs careful coordination between multiple impactful hyper-parameters.

**Number of local steps** $\tau$. Fig. 4(c) shows that as the number of local steps increases from 4 to 32, the validation accuracy decreases slightly and training loss increases. This concludes that for DaSGD, the number of local steps should be as small as possible, as long as $\tau$ is large enough to ensure parallel weight communication is totally overlapped by local computation.

**Local update percentage.** Fig. 4(d) shows that when combining the local weights with the average global weights, different percentages of local updates have little impact on validation accuracy. From the update rule equation 4, the local update proportion shares the same meaning as the momentum in hyper-parameters. One cycle policy in Fast.Ai has shown that different momentum has little effect on validation accuracy (Howard, 2018).

**Delay factor** $d$. Fig. 4(e) tunes the delay factor from 0 to 7 by fixing the number of local steps as 8, since the delay factor is bounded by the number of local steps. Results show that the delay factor imposes little effect on the convergence rate. When the delay factor increases from 0 to 7, the convergence rate slows down and the validation accuracy slightly decreases. Local SGD is just as the delay factor of 0. Thus, DaSGD incurs slightly worse validation accuracy when using the delay factor of 1, with the same local steps.

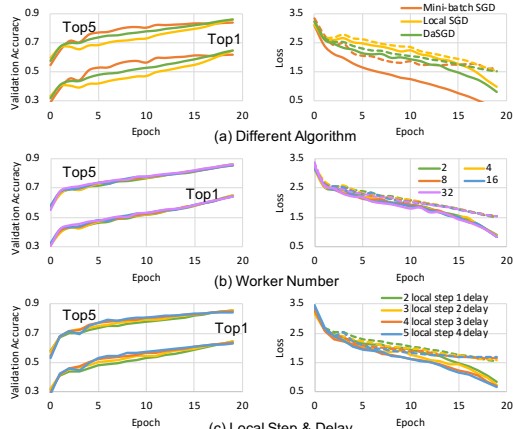

Figure 4: Effect of different parameters of DaSGD based on ResNet-50. The value in legend is the validation accuracy after 50 epochs.

Figure 5: Top-1/top-5 validation accuracy (left), training loss (right) and validation loss (right, dotted line) based on ImageNet dataset.

### 4.4 IMAGENET

We use Fast.Ai framework for training on ImageNet. The learning rate increases linearly from $0.0005$ to $0.005$ in the first $20\%$ epochs and decreases linearly from $0.005$ to $0.00005$ in the last $80\%$ epochs. The weight decay is set as $0.01$, and the momentum of SGD is $0.9$.

As shown in Fig. 5(a), three algorithms, which are Mini-batch SGD, Local SGD, and DaSGD, are trained in 20 epochs to compare the convergence rate and top-1/top-5 validation accuracy in ResNet-18. All models are trained on 8 workers and the batch size is 256. For Local SGD and DaSGD, the number of local steps $\tau$ is 2. For DaSGD, the delayed update steps $d$ is 1. The top-1/top-5 validation accuracy of DaSGD (top-1 accuracy is $0.6461$ and top-5 accuracy is $0.8582$) is the same as that of Local SGD after 20 epochs, and it is better than that of Mini-batch SGD. Although at the beginning of training, the convergence rate of DaSGD is slower than that of Mini-batch SGD, after 15 epochs, it is much faster. We also evaluate the effect of worker number in terms of the training/validation loss and top-1/top-5 validation accuracy for the ResNet-18 model based on DaSGD in Fig. 5(b). It can be found that the increase of worker number has almost no effect on the validation accuracy and convergence rate, which also reflects the application potential of the DaSGD algorithm in large-scale distributed training. Fig. 5(c) shows the results of DaSGD with different local steps and different delays. The larger local steps and the larger delay converge faster at the beginning of training, but the final accuracy is slightly poor.

## 5 CONCLUSION

In this work, we propose a new SGD algorithm called DaSGD, which parallelizes SGD and forward/back propagation to hide communication time. Just adjusting the update schedule at the software level, the DaSGD algorithm makes better use of distributed training systems and reduces the reliance on low latency and high peak throughput communication hardware. Theoretical analysis and experimental results clarify that its convergence rate is $O\left(1/\sqrt{K}\right)$, which is the same as the mini-batch SGD.

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

## A  APPENDIX

### CONVERGENCE ANALYSIS OF DASGD

#### A.1  ASSUMPTIONS

We define some notations. $\mathcal{S}$ is the training dataset, $\mathcal{S}_k$ is set $\left\{ s_k^{(1)}, ..., s_k^{(M)} \right\}$ of randomly sampled local batches at $M$ workers in $k$ iteration, $L$ is the Lipschitz constant, $d$ is the number of local iteration that global weight updates are delayed, $\tau$ is the number of local steps, $x$ is the weight of devices. The convergence analysis is conducted under the following assumptions:

- Lipschitzian gradient: $|| \bigtriangledown F(x) - \bigtriangledown F(y)|| \leq L||x - y||$
- Unbiased gradients: $E_{\mathcal{S}_k|x} [g(x)] = \bigtriangledown F(x)$
- Lower bounder: $F(x) \geq F_{inf}$
- Bounded variance: $E_{\mathcal{S}_k|x}||g(x) - \bigtriangledown F(x)||^2 \leq \sigma^2$
- Independence: All random variables are independent to each other
- Bounded age: The delay is bounded, $d \leq \tau$

#### A.2  UPDATE RULE

The update rule of DaSGD is given by

$$x_{k+1}^{(m)} = \begin{cases} x_k^{(m)} - \eta g\left(x_k^{(m)}\right), & \text{otherwise} \\ \xi x_k^{(m)} - \eta \xi g\left(x_k^{(m)}\right) + \frac{1-\xi}{M} \sum_{j=1}^{M}\left[x_{k-d}^{(j)} - \eta g\left(x_{k-d}^{(j)}\right)\right], & (k+1-d) \mod \tau = 0 \end{cases}$$

where $x_k^{(m)}$ is the weights at $m$ worker in $k$ iteration, $\eta$ is the learning rate, $M$ is the number of workers, $g(x_k^{(m)})$ is the stochastic gradient of worker $m$, $\xi$ is the local update proportion, delayed update is the case $(k+1-d) \mod \tau = 0$.

**Matrix Representation.** Define matrices $\boldsymbol{X}_k, \boldsymbol{G}_k \in \mathbb{R}^{d \times M}$ that concatenate all local models and gradients in $k$ iteration:

$$\boldsymbol{X}_k = \left[x_k^1, ..., x_k^m\right], \ \boldsymbol{G}_k = \left[g\left(x_k^{(1)}\right), ..., g\left(x_k^{(m)}\right)\right]$$

Then, the update rule is

$$\boldsymbol{X}_{k+1} = \begin{cases} \xi\left(\boldsymbol{X}_k - \eta \boldsymbol{G}_k\right) + (1-\xi)\left(\boldsymbol{X}_{k-d} - \eta \boldsymbol{G}_{k-d}\right)\boldsymbol{J}, & (k+1-d) \mod \tau = 0 \\ \boldsymbol{X}_k - \eta \boldsymbol{G}_k, & \text{otherwise} \end{cases} \quad (5)$$

**Update Rule for the Averaged Model.** The update rule of DaSGD is given by

$$x_{k+1}^{(m)} = \begin{cases} x_k^{(m)} - \eta g\left(x_k^{(m)}\right), & \text{otherwise} \\ \xi x_k^{(m)} - \eta \xi g\left(x_k^{(m)}\right) + \frac{1-\xi}{M} \sum_{j=1}^{M}\left[x_{k-d}^{(j)} - \eta g\left(x_{k-d}^{(j)}\right)\right], & (k+1-d) \mod \tau = 0 \end{cases}$$

Here, we set

$$\bar{x}_k = \frac{1}{M} \sum_{i=1}^{M} x_k^{(i)}, \ \bar{g}_k = \frac{1}{M} \sum_{i=1}^{M} g\left(x_k^{(i)}\right)$$

The average weight on different workers is obtained by

$$
\bar{x}_{k+1} = \begin{cases} \bar{x}_k - \eta \bar{g}_k, & \text{otherwise} \\ \xi \bar{x}_k + (1-\xi)\bar{x}_{k-d} - \eta \xi \bar{g}_k - \eta(1-\xi)\bar{g}(x_{k-d}), & (k+1-d) \mod \tau = 0 \end{cases}
$$

When $z = \tau(k+1)$ for $z \mod \tau = 0$, we have

$$
\begin{aligned}
\bar{x}_{\tau(k+1)+d} &= \xi \bar{x}_{\tau(k+1)+d-1} + (1-\xi)\bar{x}_{\tau(k+1)-1} - \xi\eta \bar{g}_{\tau(k+1)+d-1} - (1-\xi)\eta \bar{g}_{\tau(k+1)-1} \\
&= \xi \bar{x}_{\tau k+d} + (1-\xi)\bar{x}_{\tau k+d} - \xi\eta \sum_{i=0}^{\tau-1} \bar{g}_{\tau k+d+i} - (1-\xi)\eta \sum_{i=0}^{\tau-1-d} \bar{g}_{\tau k+d+i} \\
&= \bar{x}_{\tau k+d} - \eta \left[ \xi \left( \sum_{i=0}^{\tau-1} \bar{g}_{\tau k+d+i} - \sum_{i=0}^{\tau-1-d} \bar{g}_{\tau k+d+i} \right) + \sum_{i=0}^{\tau-1-d} \bar{g}_{\tau k+d+i} \right] \\
&= \bar{x}_{\tau k+d} - \eta \left[ \xi \sum_{i=\tau-d}^{\tau-1} \bar{g}_{\tau k+d+i} + \sum_{i=0}^{\tau-1-d} \bar{g}_{\tau k+d+i} \right]
\end{aligned}
$$

If we set $K(k) = \tau k + d$

$$
\bar{x}_{K(k+1)} = \bar{x}_{K(k)} - \eta \left[ \xi \sum_{i=\tau-d}^{\tau-1} \bar{g}_{K(k)+i} + \sum_{i=0}^{\tau-1-d} \bar{g}_{K(k)+i} \right]
$$

For the ease of writing, we first define some notations. Let $\mathcal{S}_k$ denote the set $\left\{ s_k^{(1)}, ..., s_k^{(m)} \right\}$ of mini-batches at $m$ workers in iteration $k$. Besides, define averaged stochastic gradient and averaged full batch gradient as follows:

$$
\mathcal{G}_{K(k)} = \frac{1}{M} \sum_{m=1}^{M} \left[ \sum_{i=\tau-d}^{\tau-1} \xi g\left(x_{\tau k+d+i}^{(m)}\right) + \sum_{i=0}^{\tau-1-d} g\left(x_{\tau k+d+i}^{(m)}\right) \right] \tag{6}
$$

$$
\mathcal{H}_{K(k)} = \frac{1}{M} \sum_{m=1}^{M} \left[ \sum_{i=\tau-d}^{\tau-1} \xi \nabla F\left(x_{\tau k+d+i}^{(m)}\right) + \sum_{i=0}^{\tau-1-d} \nabla F\left(x_{\tau k+d+i}^{(m)}\right) \right] \tag{7}
$$

$$
\mu_{K(k)} = \frac{1}{M} \sum_{i=1}^{M} x_{\tau k+d}^{(i)} \tag{8}
$$

Then we have

$$
\mu_{K(k+1)} = \mu_{K(k)} - \eta \mathcal{G}_{K(k)}
$$

### A.3 CONVERGENCE RATE

**Theorem (Convergence of DaSGD).** When the learning rate satisfies the following two formulas at the same time

$$
2L\eta d\xi^2 - \xi + \frac{6\xi L^2 \eta^2 d + 6L^2 \eta^2(\tau-d)}{1-\xi^2} + 6\xi L^2 \eta^2 d \le 0
$$

$$
2L\eta(\tau-d) - \xi + \frac{6\xi L^2 \eta^2 d + 6L^2 \eta^2(\tau-d)}{1-\xi^2} + 6\xi L^2 \eta^2 d + 6L^2 \eta^2(\tau-d) \le 0
$$

Then the average-squared gradient norm after $K$ iterations is bounded as

$$
\mathbb{E}_{K(k)} \left[ \frac{1}{K} \sum_{k=1}^{K} \left\| \nabla F(\mu_{K(k)}) \right\|^2 \right]
$$

$$
\le \frac{2\left[F(\mu_1) - F_{inf}\right]}{\eta K(\xi d + \tau - d)} + \frac{\eta 2L\sigma^2 \left[\xi^2 d + \tau - d\right]}{M(\xi d + \tau - d)} + \eta^2 \frac{6L^2(1+\xi)}{\xi d + \tau - d} \sum_{l=\tau-d}^{\tau-1} \sum_{i=0}^{l} \sigma^2 + \eta^2 \frac{6d\xi^2 L^2 \tau \sigma^2(1+\xi)}{(\xi d + \tau - d)(1-\xi^2)}
$$

$$
+ \frac{\eta^2}{K} \frac{12d\sigma^2 L^2 \xi^2 (\tau-d)(1+\xi)}{(\xi d + \tau - d)(1-\xi^2)} + \frac{\eta^2}{KM} \frac{12L^2 d\xi^2(\tau-d)(1+\xi)}{(\xi d + \tau - d)(1-\xi^2)} \sum_{i=1}^{d-1} \left\| \nabla F(\mathbf{X}_i) \right\|_F^2
$$

where $\mu_k = \frac{1}{M}\sum_{i=1}^{M} x_{\tau k+d}^{(i)}$, $\|\ \|_F^2$ is the Frobenius norm.

**Corollary.** Under sumptions, if the learning rate is $\eta = \frac{M+V}{M}\sqrt{\frac{M}{K}}$ the average-squared gradient norm after $K$ iterations is bounded by

$$\mathbb{E}_{K(k)}\left[\frac{1}{K}\sum_{k=1}^{K}\left\|\nabla F(\mu_{K(k)})\right\|^2\right]$$

$$\leq \frac{2\left[F(\mu_1)-F_{inf}\right]}{\sqrt{MK}(\xi d+\tau-d)} + \frac{1}{\sqrt{MK}}\frac{2L\sigma^2\left[\xi^2 d+\tau-d\right]}{(\xi d+\tau-d)}$$

$$+ \frac{M^2}{K^3}\left(1+\frac{V}{M}\right)^4\frac{6L^2(1+\xi)}{\xi d+\tau-d}\sum_{l=\tau-d}^{\tau-1}\sum_{i=0}^{l}\sigma^2 + \frac{M^2}{K^3}\left(1+\frac{V}{M}\right)^4\frac{6d\xi^2 L^2\tau\sigma^2(1+\xi)}{(\xi d+\tau-d)(1-\xi^2)}$$

$$+ \frac{M^2}{K^3}\left(1+\frac{V}{M}\right)^4\frac{12d\sigma^2 L^2\xi^2(\tau-d)(1+\xi)}{(\xi d+\tau-d)(1-\xi^2)} + \frac{M}{K^3}\left(1+\frac{V}{M}\right)^4\frac{12L^2 d\xi^2(\tau-d)(1+\xi)}{(\xi d+\tau-d)(1-\xi^2)}\sum_{i=1}^{d-1}\|\nabla F(\mathbf{X}_i)\|_F^2$$

If the total iterations $K$ is sufficiently large, then the average-squared gradient norm will be bounded by

$$\mathbb{E}\left[\frac{1}{K}\sum_{k=1}^{K}\|\nabla F(\mu_k)\|^2\right] \leq \frac{2\left[F(\mu_1)-F_{inf}\right]+2L\sigma^2\left[\xi^2 d+\tau-d\right]}{\sqrt{MK}(\xi d+\tau-d)}$$

.

## A.4 PROOF OF CONVERGENCE RATE

**Lemma 1.** If the learning rate satisfies $\eta \leq \min\left\{\frac{1}{2Ld\xi}, \frac{\xi}{2L(\tau-d)}\right\}$ and all local model parameters are initialized at the same point, then the average-squared gradient after $K$ iterations is bounded as follows

$$\mathbb{E}_{K(k)}\left[\frac{1}{K}\sum_{k=1}^{K}\left\|\nabla F(\mu_{K(k)})\right\|^2\right]$$

$$\leq \frac{2\left[F(\mu_1)-F_{inf}\right]}{\eta K(\xi d+\tau-d)} + \frac{2L\eta\sigma^2\left[\xi^2 d+\tau-d\right]}{M(\xi d+\tau-d)}$$

$$+ \frac{L^2}{KM(\xi d+\tau-d)}\sum_{k=1}^{K}\sum_{m=1}^{M}\left[\sum_{i=0}^{\tau-1-d}\mathbb{E}_{K(k)}\left\|\mu_{K(k)}-x_{\tau k+d+i}^{(m)}\right\|^2 + \xi\sum_{i=\tau-d}^{\tau-1}\mathbb{E}_{K(k)}\left\|\mu_{K(k)}-x_{\tau k+d+i}^{(m)}\right\|^2\right]$$

Proof.

From the Lipschitzisan gradient assumption $\|\nabla F(x)-\nabla F(y)\| \leq L\|x-y\|$, we have

$$F(X_{K(k+1)}) - F(X_{K(k)}) \leq \left\langle\nabla F(X_{K(k)}), X_{K(k+1)}-X_{K(k)}\right\rangle + \frac{L}{2}\left\|X_{K(k+1)}-X_{K(k)}\right\|^2$$

$$= -\eta\left\langle\nabla F(X_{K(k)}), \mathcal{G}_{K(k)}\right\rangle + \frac{L\eta^2}{2}\left\|\mathcal{G}_{K(k)}\right\|^2 \tag{9}$$

Taking expectation respect to $\mathcal{S}_{K(k)}$ on both sides of equation 9, we have

$$\mathbb{E}_{K(k)}\left[F(X_{K(k+1)})\right] - F(X_{K(k)}) \leq -\eta\mathbb{E}_{K(k)}\left[\left\langle\nabla F(X_{K(k)}), \mathcal{G}_{K(k)}\right\rangle\right] + \frac{L\eta^2}{2}\mathbb{E}_{K(k)}\left[\left\|\mathcal{G}_{K(k)}\right\|^2\right]$$

From the fact

$$\langle a,b\rangle = \frac{1}{2}\left(\|a\|^2+\|b\|^2-\|a-b\|^2\right)$$

we have

$$\mathbb{E}_{K(k)}\left[F(X_{K(k+1)})\right] - F(X_{K(k)}) \leq -\eta\mathbb{E}_{K(k)}\left[\langle\triangledown F(X_{K(k)}),\mathcal{G}_{K(k)}\rangle\right] + \frac{L\eta^2}{2}\mathbb{E}_{K(k)}\left[\left\|\mathcal{G}_{K(k)}\right\|^2\right]$$

Combining with Lemmas 4 and 5, we obtain

$$\mathbb{E}_{K(k)}\left[F(X_{K(k+1)})\right] - F(X_{K(k)})$$

$$\leq -\eta\mathbb{E}_{K(k)}\left[\langle\triangledown F(X_{K(k)}),\mathcal{G}_{K(k)}\rangle\right] + \frac{L\eta^2}{2}\mathbb{E}_{K(k)}\left[\left\|\mathcal{G}_{K(k)}\right\|^2\right]$$

$$\leq -\eta\frac{\xi d + \tau - d}{2}\left\|\triangledown F(X_{K(k)})\right\|^2 + \frac{L\eta^2\sigma^2}{M}\left[d\xi^2 + \tau - d\right]$$

$$+ \left[\frac{L\eta^2 d\xi^2}{M} - \frac{\eta\xi}{2M}\right]\sum_{i=\tau-d}^{\tau-1}\left\|\triangledown F\left(\mathbf{X}_{\tau k+d+i}\right)\right\|_F^2 + \left[\frac{L\eta^2(\tau - d)}{M} - \frac{\eta\xi}{2M}\right]\sum_{i=0}^{\tau-1-d}\left\|\triangledown F\left(\mathbf{X}_{\tau k+d+i}\right)\right\|_F^2$$

$$+ \frac{\eta}{2M}\sum_{m=1}^{M}\left[\sum_{i=0}^{\tau-1-d}\left\|\triangledown F(X_{K(k)}) - \triangledown F\left(x_{\tau k+d+i}^{(m)}\right)\right\|^2 + \xi\sum_{i=\tau-d}^{\tau-1}\left\|\triangledown F(X_{K(k)}) - \triangledown F\left(x_{\tau k+d+i}^{(m)}\right)\right\|^2\right]$$

$$\leq -\eta\frac{\xi d + \tau - d}{2}\left\|\triangledown F(\mu_{K(k)})\right\|^2 + \frac{L\eta^2\sigma^2}{M}\left[d\xi^2 + \tau - d\right]$$

$$+ \left[\frac{L\eta^2 d\xi^2}{M} - \frac{\eta\xi}{2M}\right]\sum_{i=\tau-d}^{\tau-1}\left\|\triangledown F\left(\mathbf{X}_{\tau k+d+i}\right)\right\|_F^2 + \left[\frac{L\eta^2(\tau - d)}{M} - \frac{\eta\xi}{2M}\right]\sum_{i=0}^{\tau-1-d}\left\|\triangledown F\left(\mathbf{X}_{\tau k+d+i}\right)\right\|_F^2$$

$$+ \frac{\eta L^2}{2M}\sum_{m=1}^{M}\left[\sum_{i=0}^{\tau-1-d}\left\|\mu_{K(k)} - x_{\tau k+d+i}^{(m)}\right\|^2 + \xi\sum_{i=\tau-d}^{\tau-1}\left\|\mu_{K(k)} - x_{\tau k+d+i}^{(m)}\right\|^2\right] \qquad (10)$$

where equation 10 is due to the Lipschitzisan gradient assumption $||\triangledown F(x) - \triangledown F(y)|| \leq L||x-y||$. After minor rearranging and according to the definition of Frobenius norm, it is easy to show

$$\eta\frac{\xi d + \tau - d}{2}\left\|\triangledown F(\mu_{K(k)})\right\|^2$$

$$\leq F(\mu_{K(k)}) - \mathbb{E}_{K(k)}\left[F(\mu_{K(k+1)})\right] + \frac{L\eta^2\sigma^2}{M}\left[d\xi^2 + \tau - d\right]$$

$$+ \left[\frac{L\eta^2 d\xi^2}{M} - \frac{\eta\xi}{2M}\right]\sum_{i=\tau-d}^{\tau-1}\left\|\triangledown F\left(\mathbf{X}_{\tau k+d+i}\right)\right\|_F^2 + \left[\frac{L\eta^2(\tau - d)}{M} - \frac{\eta\xi}{2M}\right]\sum_{i=0}^{\tau-1-d}\left\|\triangledown F\left(\mathbf{X}_{\tau k+d+i}\right)\right\|_F^2$$

$$+ \frac{\eta L^2}{2M}\sum_{m=1}^{M}\left[\sum_{i=0}^{\tau-1-d}\left\|\mu_{K(k)} - x_{\tau k+d+i}^{(m)}\right\|^2 + \xi\sum_{i=\tau-d}^{\tau-1}\left\|\mu_{K(k)} - x_{\tau k+d+i}^{(m)}\right\|^2\right]$$

Taking the total expectation and averaging over all iterates, we have

$$\eta\frac{\xi d + \tau - d}{2}\mathbb{E}_{K(k)}\left[\frac{1}{K}\sum_{k=1}^{K}\left\|\triangledown F(\mu_{K(k)})\right\|^2\right]$$

$$\leq \frac{F(\mu_1) - F_{inf}}{K} + \frac{L\eta^2\sigma^2}{M}\left[d\xi^2 + \tau - d\right]$$

$$+ \left[\frac{L\eta^2 d\xi^2}{KM} - \frac{\eta\xi}{2KM}\right]\sum_{k=1}^{K}\sum_{i=\tau-d}^{\tau-1}\left\|\triangledown F\left(\mathbf{X}_{\tau k+d+i}\right)\right\|_F^2 + \left[\frac{L\eta^2(\tau - d)}{KM} - \frac{\eta\xi}{2KM}\right]\sum_{k=1}^{K}\sum_{i=0}^{\tau-1-d}\left\|\triangledown F\left(\mathbf{X}_{\tau k+d+i}\right)\right\|_F^2$$

$$+ \frac{\eta L^2}{2KM}\sum_{k=1}^{K}\sum_{m=1}^{M}\left[\sum_{i=0}^{\tau-1-d}\left\|\mu_{K(k)} - x_{\tau k+d+i}^{(m)}\right\|^2 + \xi\sum_{i=\tau-d}^{\tau-1}\left\|\mu_{K(k)} - x_{\tau k+d+i}^{(m)}\right\|^2\right]$$

Then, we have

$$
\begin{aligned}
\mathbb{E}_{K(k)} \left[ \frac{1}{K} \sum_{k=1}^{K} \left\| \nabla F(\mu_{K(k)}) \right\|^2 \right] \leq & \frac{2 \left[ F(\mu_1) - F_{inf} \right]}{\eta K(\xi d + \tau - d)} + \frac{2L\eta\sigma^2 \left[ \xi^2 d + \tau - d \right]}{M(\xi d + \tau - d)} \\
& + \frac{2L\eta d\xi^2 - \xi}{KM(\xi d + \tau - d)} \sum_{k=1}^{K} \sum_{i=\tau-d}^{\tau-1} \mathbb{E}_{K(k)} \left\| \nabla F\left( \mathbf{X}_{\tau k+d+i} \right) \right\|_F^2 \\
& + \frac{2L\eta(\tau - d) - \xi}{KM(\xi d + \tau - d)} \sum_{k=1}^{K} \sum_{i=0}^{\tau-1-d} \mathbb{E}_{K(k)} \left\| \nabla F\left( \mathbf{X}_{\tau k+d+i} \right) \right\|_F^2 \\
& + \frac{L^2}{KM(\xi d + \tau - d)} \sum_{k=1}^{K} \sum_{i=0}^{\tau-1-d} \sum_{m=1}^{M} \mathbb{E}_{K(k)} \left\| \mu_{K(k)} - x_{\tau k+d+i}^{(m)} \right\|^2 \\
& + \frac{\xi L^2}{KM(\xi d + \tau - d)} \sum_{k=1}^{K} \sum_{i=\tau-d}^{\tau-1} \sum_{m=1}^{M} \mathbb{E}_{K(k)} \left\| \mu_{K(k)} - x_{\tau k+d+i}^{(m)} \right\|^2
\end{aligned}
\tag{11}
$$

If the learning rate satisfies $\eta \leq \min\left\{ \frac{1}{2Ld\xi}, \frac{\xi}{2L(\tau-d)} \right\}$, then

$$
\begin{aligned}
\mathbb{E}_{K(k)} \left[ \frac{1}{K} \sum_{k=1}^{K} \left\| \nabla F(\mu_{K(k)}) \right\|^2 \right] \leq & \frac{2 \left[ F(\mu_1) - F_{inf} \right]}{\eta K(\xi d + \tau - d)} + \frac{2L\eta\sigma^2 \left[ \xi^2 d + \tau - d \right]}{M(\xi d + \tau - d)} \\
& + \frac{L^2}{KM(\xi d + \tau - d)} \sum_{k=1}^{K} \sum_{i=0}^{\tau-1-d} \sum_{m=1}^{M} \mathbb{E}_{K(k)} \left\| \mu_{K(k)} - x_{\tau k+d+i}^{(m)} \right\|^2 \\
& + \frac{\xi L^2}{KM(\xi d + \tau - d)} \sum_{k=1}^{K} \sum_{i=\tau-d}^{\tau-1} \sum_{m=1}^{M} \mathbb{E}_{K(k)} \left\| \mu_{K(k)} - x_{\tau k+d+i}^{(m)} \right\|^2
\end{aligned}
$$

Recalling the definition $\mu_{K(k)} = \frac{1}{M} \sum_{i=1}^{M} x_{\tau k+d}^{(i)} = \mathbf{X}_{K(k)} \mathbf{1}_M / M$ and adding a positive term to the RHS, one can get

$$
\sum_{i=\tau-d}^{\tau-1} \sum_{m=1}^{M} \left\| \mu_{K(k)} - x_{\tau k+d+i}^{(m)} \right\|^2 = \sum_{i=\tau-d}^{\tau-1} \left\| \mathbf{X}_{\tau k+d} \mathbf{J} - \mathbf{X}_{\tau k+d+i} \right\|_F^2
$$

We have

$$
\begin{aligned}
\mathbb{E}_{K(k)} \left[ \frac{1}{K} \sum_{k=1}^{K} \left\| \nabla F(\mu_{K(k)}) \right\|^2 \right] \leq & \frac{2 \left[ F(\mu_1) - F_{inf} \right]}{\eta K(\xi d + \tau - d)} + \frac{2L\eta\sigma^2 \left[ \xi^2 d + \tau - d \right]}{M(\xi d + \tau - d)} \\
& + \frac{L^2}{KM(\xi d + \tau - d)} \sum_{k=1}^{K} \sum_{i=0}^{\tau-1-d} \mathbb{E}_{K(k)} \left\| \mathbf{X}_{\tau k+d} \mathbf{J} - \mathbf{X}_{\tau k+d+i} \right\|_F^2 \\
& + \frac{\xi L^2}{KM(\xi d + \tau - d)} \sum_{k=1}^{K} \sum_{i=\tau-d}^{\tau-1} \mathbb{E}_{K(k)} \left\| \mathbf{X}_{\tau k+d} \mathbf{J} - \mathbf{X}_{\tau k+d+i} \right\|_F^2
\end{aligned}
$$

**Lemma 2.**

$$
\left\| \mathcal{H}_{K(k)} \right\|^2 \leq \frac{2d\xi^2}{M} \sum_{i=\tau-d}^{\tau-1} \left\| \nabla F\left( \mathbf{X}_{\tau k+d+i} \right) \right\|_F^2 + \frac{2(\tau - d)}{M} \sum_{i=0}^{\tau-1-d} \left\| \nabla F\left( \mathbf{X}_{\tau k+d+i} \right) \right\|_F^2
\tag{12}
$$

Proof.

$$\left\| \mathcal{H}_{K(k)} \right\|^2 = \left\| \xi \frac{1}{M} \sum_{i=\tau-d}^{\tau-1} \sum_{m=1}^{M} \nabla F\left(x_{\tau k+d+i}^{(m)}\right) + \frac{1}{M} \sum_{i=0}^{\tau-1-d} \sum_{m=1}^{M} \nabla F\left(x_{\tau k+d+i}^{(m)}\right) \right\|^2$$

$$\leq \frac{2d\xi^2}{M^2} \sum_{i=\tau-d}^{\tau-1} \left\| \sum_{m=1}^{M} \nabla F\left(x_{\tau k+d+i}^{(m)}\right) \right\|^2 + \frac{2(\tau-d)}{M^2} \sum_{i=0}^{\tau-1-d} \left\| \sum_{m=1}^{M} \nabla F\left(x_{\tau k+d+i}^{(m)}\right) \right\|^2 \tag{13}$$

$$\leq \frac{2d\xi^2}{M} \sum_{i=\tau-d}^{\tau-1} \sum_{m=1}^{M} \left\| \nabla F\left(x_{\tau k+d+i}^{(m)}\right) \right\|^2 + \frac{2(\tau-d)}{M} \sum_{i=0}^{\tau-1-d} \sum_{m=1}^{M} \left\| \nabla F\left(x_{\tau k+d+i}^{(m)}\right) \right\|^2 \tag{14}$$

$$= \frac{2d\xi^2}{M} \sum_{i=\tau-d}^{\tau-1} \left\| \nabla F\left(\mathbf{X}_{\tau k+d+i}\right) \right\|_F^2 + \frac{2(\tau-d)}{M} \sum_{i=0}^{\tau-1-d} \left\| \nabla F\left(\mathbf{X}_{\tau k+d+i}\right) \right\|_F^2$$

where equation 13 is due to $\|a+b\|^2 \leq 2\|a\|^2 + 2\|b\|^2$, equation 14 comes from the convexity of vector norm and Jensen's inequality.

---

**Lemma 3.** Under assumptions $\mathbb{E}_{\mathcal{S}_k|x}[g(x)] = \nabla F(x)$ and $\mathbb{E}_{\mathcal{S}_k|x}\|g(x)-\nabla F(x)\|^2 \leq \sigma^2$, we have the following variance bound for the averaged stochastic gradient:

$$\mathbb{E}_{K(k)}\left[\left\|\mathcal{G}_{K(k)} - \mathcal{H}_{K(k)}\right\|^2\right] \leq \frac{2\sigma^2}{M}\left[d\xi^2 + \tau - d\right] \tag{15}$$

Proof. According to the definition of equation 6, equation 7, and equation 8, we have

$$\mathbb{E}_{K(k)}\left[\left\|\mathcal{G}_{K(k)} - \mathcal{H}_{K(k)}\right\|^2\right]$$

$$= \frac{1}{M^2}\mathbb{E}_{K(k)}\left[\left\| \xi \sum_{i=\tau-d}^{\tau-1} \sum_{m=1}^{M} \left[g\left(x_{\tau k+d+i}^{(m)}\right) - \nabla F\left(x_{\tau k+d+i}^{(m)}\right)\right] + \sum_{i=0}^{\tau-1-d} \sum_{m=1}^{M} \left[g\left(x_{\tau k+d+i}^{(m)}\right) - \nabla F\left(x_{\tau k+d+i}^{(m)}\right)\right] \right\|^2\right]$$

$$\leq \frac{2}{M^2}\mathbb{E}_{K(k)}\left[\left\| \xi \sum_{i=\tau-d}^{\tau-1} \sum_{m=1}^{M} \left[g\left(x_{\tau k+d+i}^{(m)}\right) - \nabla F\left(x_{\tau k+d+i}^{(m)}\right)\right] \right\|^2 + \left\| \sum_{i=0}^{\tau-1-d} \sum_{m=1}^{M} \left[g\left(x_{\tau k+d+i}^{(m)}\right) - \nabla F\left(x_{\tau k+d+i}^{(m)}\right)\right] \right\|^2\right] \tag{16}$$

$$= \frac{2}{M^2}\mathbb{E}_{K(k)}\left[ \xi^2 \sum_{i=\tau-d}^{\tau-1} \sum_{m=1}^{M} \left\| g\left(x_{\tau k+d+i}^{(m)}\right) - \nabla F\left(x_{\tau k+d+i}^{(m)}\right) \right\|^2 + \sum_{i=0}^{\tau-1-d} \sum_{m=1}^{M} \left\| g\left(x_{\tau k+d+i}^{(m)}\right) - \nabla F\left(x_{\tau k+d+i}^{(m)}\right) \right\|^2 \right. \tag{17}$$

$$+ \xi^2 \sum_{j\neq i}^{\tau-1} \sum_{l\neq m}^{M} \left\langle g\left(x_{\tau k+d+i}^{(m)}\right) - \nabla F\left(x_{\tau k+d+i}^{(m)}\right), g\left(x_{\tau k+d+j}^{(l)}\right) - \nabla F\left(x_{\tau k+d+j}^{(l)}\right) \right\rangle \tag{18}$$

$$\left. + \sum_{j\neq i}^{\tau-1-d} \sum_{l\neq m}^{M} \left\langle g\left(x_{\tau k+d+i}^{(m)}\right) - \nabla F\left(x_{\tau k+d+i}^{(m)}\right), g\left(x_{\tau k+d+j}^{(l)}\right) - \nabla F\left(x_{\tau k+d+j}^{(l)}\right) \right\rangle \right] \tag{19}$$

$$= \frac{2\xi^2}{M^2} \sum_{i=\tau-d}^{\tau-1} \sum_{m=1}^{M} \mathbb{E}_{K(k)} \left\| g\left(x_{\tau k+d+i}^{(m)}\right) - \nabla F\left(x_{\tau k+d+i}^{(m)}\right) \right\|^2 + \frac{2}{M^2} \sum_{i=0}^{\tau-1-d} \sum_{m=1}^{M} \mathbb{E}_{K(k)} \left\| g\left(x_{\tau k+d+i}^{(m)}\right) - \nabla F\left(x_{\tau k+d+i}^{(m)}\right) \right\|^2 \tag{20}$$

where equation 16 is due to $\|a+b\|^2 \leq 2\|a\|^2 + 2\|b\|^2$, equation 20 is due to $s_k^i$ are independent random variables and the assumption $\mathbb{E}_{\mathcal{S}_k|x}[g(x)] = \nabla F(x)$. Now, directly applying assumption

$\mathbb{E}_{\mathcal{S}_k|x}||g(x) - \triangledown F(x)||^2 \leq \sigma^2$ to equation 20. Then, we have

$$\mathbb{E}_{K(k)}\left[\left\|\mathcal{G}_{K(k)} - \mathcal{H}_{K(k)}\right\|^2\right] \leq \frac{2\xi^2}{M^2} \sum_{i=\tau-d}^{\tau-1} \sum_{m=1}^{M} \sigma^2 + \frac{2}{M^2} \sum_{i=0}^{\tau-1-d} \sum_{m=1}^{M} \sigma^2 = \frac{2\sigma^2}{M}\left[d\xi^2 + \tau - d\right]$$

**Lemma 4.** Under assumption $\mathbb{E}_{\mathcal{S}_k|x}\left[g(x)\right] = \triangledown F(x)$, the expected inner product between stochastic gradient and full batch gradient can be expanded as

$$\mathbb{E}_{K(k)}\left[\left\langle \triangledown F(X_{K(k)}), \mathcal{G}_{K(k)} \right\rangle\right]$$
$$= \frac{\xi d + \tau - d}{2}\left\|\triangledown F(X_{K(k)})\right\|^2 + \frac{1}{2M}\left[\xi \sum_{i=\tau-d}^{\tau-1}\left\|\triangledown F\left(\mathbf{X}_{\tau k+d+i}\right)\right\|_F^2 + \sum_{i=0}^{\tau-1-d}\left\|\triangledown F\left(\mathbf{X}_{\tau k+d+i}\right)\right\|_F^2\right]$$
$$- \frac{1}{2M}\sum_{m=1}^{M}\left[\sum_{i=0}^{\tau-1-d}\left\|\triangledown F(X_{K(k)}) - \triangledown F\left(x_{\tau k+d+i}^{(m)}\right)\right\|^2 + \xi \sum_{i=\tau-d}^{\tau-1}\left\|\triangledown F(X_{K(k)}) - \triangledown F\left(x_{\tau k+d+i}^{(m)}\right)\right\|^2\right]$$

Proof.

$$\mathbb{E}_{K(k)}\left[\left\langle \triangledown F(X_{K(k)}), \mathcal{G}_{K(k)} \right\rangle\right]$$
$$= \mathbb{E}_{K(k)}\left[\left\langle \triangledown F(X_{K(k)}), \xi\frac{1}{M}\sum_{i=\tau-d}^{\tau-1}\sum_{m=1}^{M} g\left(x_{\tau k+d+i}^{(m)}\right) + \frac{1}{M}\sum_{i=0}^{\tau-1-d}\sum_{m=1}^{M} g\left(x_{\tau k+d+i}^{(m)}\right)\right\rangle\right]$$
$$= \xi\frac{1}{M}\sum_{i=\tau-d}^{\tau-1}\sum_{m=1}^{M}\left\langle \triangledown F(X_{K(k)}), \triangledown F\left(x_{\tau k+d+i}^{(m)}\right)\right\rangle + \frac{1}{M}\sum_{i=0}^{\tau-1-d}\sum_{m=1}^{M}\left\langle \triangledown F(X_{K(k)}), \triangledown F\left(x_{\tau k+d+i}^{(m)}\right)\right\rangle$$
$$= \frac{\xi}{2M}\sum_{i=\tau-d}^{\tau-1}\sum_{m=1}^{M}\left[\left\|\triangledown F(X_{K(k)})\right\|^2 + \left\|\triangledown F\left(x_{\tau k+d+i}^{(m)}\right)\right\|^2 - \left\|\triangledown F(X_{K(k)}) - \triangledown F\left(x_{\tau k+d+i}^{(m)}\right)\right\|^2\right]$$
$$\tag{21}$$

$$+ \frac{1}{2M}\sum_{i=0}^{\tau-1-d}\sum_{m=1}^{M}\left[\left\|\triangledown F(X_{K(k)})\right\|^2 + \left\|\triangledown F\left(x_{\tau k+d+i}^{(m)}\right)\right\|^2 - \left\|\triangledown F(X_{K(k)}) - \triangledown F\left(x_{\tau k+d+i}^{(m)}\right)\right\|^2\right]$$
$$\tag{22}$$

$$= \frac{\xi d + \tau - d}{2}\left\|\triangledown F(X_{K(k)})\right\|^2 + \frac{1}{2M}\left[\xi \sum_{i=\tau-d}^{\tau-1}\left\|\triangledown F\left(\mathbf{X}_{\tau k+d+i}\right)\right\|_F^2 + \sum_{i=0}^{\tau-1-d}\left\|\triangledown F\left(\mathbf{X}_{\tau k+d+i}\right)\right\|_F^2\right]$$
$$- \frac{1}{2M}\sum_{m=1}^{M}\left[\sum_{i=0}^{\tau-1-d}\left\|\triangledown F(X_{K(k)}) - \triangledown F\left(x_{\tau k+d+i}^{(m)}\right)\right\|^2 + \xi \sum_{i=\tau-d}^{\tau-1}\left\|\triangledown F(X_{K(k)}) - \triangledown F\left(x_{\tau k+d+i}^{(m)}\right)\right\|^2\right]$$

where equation 21 and equation 22 come from $\langle a, b\rangle = \frac{1}{2}\left(||a||^2 + ||b||^2 - ||a-b||^2\right)$.

**Lemma 5.** Under assumptions $E_{\xi|x}\left[g(x)\right] = \triangledown F(x)$ and $E_{\xi|x}||g(x) - \triangledown F(x)||^2 \leq \sigma^2$, the squared norm of stochastic gradient can be bounded as

$$\mathbb{E}_{K(k)}\left[\left\|\mathcal{G}_{K(k)}\right\|^2\right] \leq \frac{2\sigma^2}{M}\left[d\xi^2 + \tau - d\right] + \frac{2d\xi^2}{M}\sum_{i=\tau-d}^{\tau-1}\left\|\triangledown F\left(\mathbf{X}_{\tau k+d+i}\right)\right\|_F^2 + \frac{2(\tau-d)}{M}\sum_{i=0}^{\tau-1-d}\left\|\triangledown F\left(\mathbf{X}_{\tau k+d+i}\right)\right\|_F^2$$

Proof.

$$
\mathbb{E}_{K(k)}\left[\left\|\mathcal{G}_{K(k)}\right\|^2\right]
$$

$$
= \mathbb{E}_{K(k)}\left[\left\|\mathcal{G}_{K(k)} - \mathbb{E}_{K(k)}[\mathcal{G}_{K(k)}]\right\|^2\right] + \left\|\mathbb{E}_{K(k)}[\mathcal{G}_{K(k)}]\right\|^2
$$

$$
= \mathbb{E}_{K(k)}\left[\left\|\mathcal{G}_{K(k)} - \mathcal{H}_{K(k)}\right\|^2\right] + \left\|\mathcal{H}_{K(k)}\right\|^2
$$

$$
\leq \frac{2\sigma^2}{M}\left[d\xi^2 + \tau - d\right] + \frac{2d\xi^2}{M}\sum_{i=\tau-d}^{\tau-1}\left\|\bigtriangledown F\left(\mathbf{X}_{\tau k+d+i}\right)\right\|_F^2 + \frac{2(\tau-d)}{M}\sum_{i=0}^{\tau-1-d}\left\|\bigtriangledown F\left(\mathbf{X}_{\tau k+d+i}\right)\right\|_F^2
\tag{23}
$$

where equation 23 follows equation 12 and equation 15.

---

**Theorem 1 (Convergence of SGD).** Under assumptions, when the learning rate satisfies the following two formulas at the same time

$$
2L\eta d\xi^2 - \xi + \frac{6\xi L^2\eta^2 d + 6L^2\eta^2(\tau-d)}{1-\xi^2} + 6\xi L^2\eta^2 d \leq 0
$$

$$
2L\eta(\tau-d) - \xi + \frac{6\xi L^2\eta^2 d + 6L^2\eta^2(\tau-d)}{1-\xi^2} + 6\xi L^2\eta^2 d + 6L^2\eta^2(\tau-d) \leq 0
$$

Then the average-squared gradient norm after $K$ iterations is bounded as

$$
\mathbb{E}_{K(k)}\left[\frac{1}{K}\sum_{k=1}^{K}\left\|\bigtriangledown F(\mu_{K(k)})\right\|^2\right]
$$

$$
\leq \frac{2\left[F(\mu_1) - F_{inf}\right]}{\eta K(\xi d + \tau - d)} + \frac{\eta 2L\sigma^2\left[\xi^2 d + \tau - d\right]}{M(\xi d + \tau - d)} + \eta^2\frac{6L^2(1+\xi)}{\xi d + \tau - d}\sum_{l=\tau-d}^{\tau-1}\sum_{i=0}^{l}\sigma^2 + \eta^2\frac{6d\xi^2 L^2\tau\sigma^2(1+\xi)}{(\xi d + \tau - d)(1-\xi^2)}
$$

$$
+ \frac{\eta^2}{K}\frac{12d\sigma^2 L^2\xi^2(\tau-d)(1+\xi)}{(\xi d + \tau - d)(1-\xi^2)} + \frac{\eta^2}{KM}\frac{12L^2 d\xi^2(\tau-d)(1+\xi)}{(\xi d + \tau - d)(1-\xi^2)}\sum_{i=1}^{d-1}\left\|\bigtriangledown F(\mathbf{X}_i)\right\|_F^2
$$

where $\mu_k = \frac{1}{M}\sum_{i=1}^{M}x_{\tau k+d}^{(i)}$, $\|\ \|_F^2$ is the Frobenius norm.

Proof.

Recall the intermediate result equation 11 in the proof of Lemma 1:

$$
\mathbb{E}_{K(k)}\left[\frac{1}{K}\sum_{k=1}^{K}\left\|\bigtriangledown F(\mu_{K(k)})\right\|^2\right] \leq \frac{2\left[F(\mu_1) - F_{inf}\right]}{\eta K(\xi d + \tau - d)} + \frac{2L\eta\sigma^2\left[\xi^2 d + \tau - d\right]}{M(\xi d + \tau - d)}
$$

$$
+ \frac{2L\eta d\xi^2 - \xi}{KM(\xi d + \tau - d)}\sum_{k=1}^{K}\sum_{i=\tau-d}^{\tau-1}\mathbb{E}_{K(k)}\left\|\bigtriangledown F\left(\mathbf{X}_{\tau k+d+i}\right)\right\|_F^2
$$

$$
+ \frac{2L\eta(\tau-d) - \xi}{KM(\xi d + \tau - d)}\sum_{k=1}^{K}\sum_{i=0}^{\tau-1-d}\mathbb{E}_{K(k)}\left\|\bigtriangledown F\left(\mathbf{X}_{\tau k+d+i}\right)\right\|_F^2
$$

$$
+ \frac{L^2}{KM(\xi d + \tau - d)}\sum_{k=1}^{K}\sum_{i=0}^{\tau-1-d}\mathbb{E}_{K(k)}\left\|\mathbf{X}_{\tau k+d}\mathbf{J} - \mathbf{X}_{\tau k+d+i}\right\|_F^2
$$

$$
+ \frac{\xi L^2}{KM(\xi d + \tau - d)}\sum_{k=1}^{K}\sum_{i=\tau-d}^{\tau-1}\mathbb{E}_{K(k)}\left\|\mathbf{X}_{\tau k+d}\mathbf{J} - \mathbf{X}_{\tau k+d+i}\right\|_F^2
\tag{24}
$$

Our goal is to provide an upper bound for the network error term $\sum_{k=1}^{K}\sum_{i=\tau-d}^{\tau-1}\mathbb{E}_{K(k)}\left\|\mathbf{X}_{\tau k+d}\mathbf{J} - \mathbf{X}_{\tau k+d+i}\right\|_F^2$. First of all, let us derive a specific expression for

$\mathbf{X}_{\tau k+d}\mathbf{J} - \mathbf{X}_{\tau k+d+i}$. According to the update rule equation 5, one can observe that

$$\mathbf{X}_{\tau k+d}\mathbf{J} - \mathbf{X}_{\tau k+d+i}$$

$$= \mathbf{X}_{\tau k+d}(\mathbf{J} - \mathbf{I}) + \eta \sum_{j=0}^{i} \mathbf{G}_{\tau k+d+j}$$

$$= \xi\left(\mathbf{X}_{\tau k+d-1} - \eta\mathbf{G}_{\tau k+d-1}\right)(\mathbf{J}-\mathbf{I}) + (1-\xi)\left(\mathbf{X}_{\tau k} - \eta\mathbf{G}_{\tau k}\right)\mathbf{J}(\mathbf{J}-\mathbf{I}) + \eta\sum_{j=0}^{i}\mathbf{G}_{\tau k+d+j}$$

$$= \xi\mathbf{X}_{\tau(k-1)+d}(\mathbf{J}-\mathbf{I}) - \xi\eta\sum_{i=0}^{\tau-1}\mathbf{G}_{\tau(k-1)+d+i}(\mathbf{J}-\mathbf{I}) + \eta\sum_{j=0}^{i}\mathbf{G}_{\tau k+d+j}$$

$$= \xi^2\mathbf{X}_{\tau(k-2)+d}(\mathbf{J}-\mathbf{I}) - \eta\sum_{j=1}^{2}\sum_{i=0}^{\tau-1}\xi^j\mathbf{G}_{\tau(k-j)+d+i}(\mathbf{J}-\mathbf{I}) + \eta\sum_{j=0}^{i}\mathbf{G}_{\tau k+d+j}$$

$$= \xi^k\mathbf{X}_d(\mathbf{J}-\mathbf{I}) - \eta\sum_{j=1}^{k}\sum_{i=0}^{\tau-1}\xi^j\mathbf{G}_{\tau(k-j)+d+i}(\mathbf{J}-\mathbf{I}) + \eta\sum_{j=0}^{i}\mathbf{G}_{\tau k+d+j}$$

$$= \xi^k\left(\mathbf{X}_{d-1} - \eta\mathbf{G}_{d-1}\right)(\mathbf{J}-\mathbf{I}) - \eta\sum_{j=1}^{k}\sum_{i=0}^{\tau-1}\xi^j\mathbf{G}_{\tau(k-j)+d+i}(\mathbf{J}-\mathbf{I}) + \eta\sum_{j=0}^{i}\mathbf{G}_{\tau k+d+j}$$

$$= \xi^k\mathbf{X}_1(\mathbf{J}-\mathbf{I}) - \eta\xi^k\sum_{i=1}^{d-1}\mathbf{G}_i(\mathbf{J}-\mathbf{I}) - \eta\sum_{j=1}^{k}\sum_{i=0}^{\tau-1}\xi^j\mathbf{G}_{\tau(k-j)+d+i}(\mathbf{J}-\mathbf{I}) + \eta\sum_{j=0}^{i}\mathbf{G}_{\tau k+d+j}$$

$$= -\eta\xi^k\sum_{i=1}^{d-1}\mathbf{G}_i(\mathbf{J}-\mathbf{I}) - \eta\sum_{j=1}^{k}\sum_{i=0}^{\tau-1}\xi^j\mathbf{G}_{\tau(k-j)+d+i}(\mathbf{J}-\mathbf{I}) + \eta\sum_{j=0}^{i}\mathbf{G}_{\tau k+d+j} \tag{25}$$

where equation 25 follows the fact that all workers start from the same point at the beginning of each local update period.

Accordingly, we have

$$\sum_{i=\tau-d}^{\tau-1} \mathbb{E}_{K(k)} \|\mathbf{X}_{\tau k+d}\mathbf{J} - \mathbf{X}_{\tau k+d+i}\|_F^2$$

$$= \sum_{l=\tau-d}^{\tau-1} \mathbb{E}_{K(k)} \left\| -\eta\xi^k \sum_{i=1}^{d-1} \mathbf{G}_i(\mathbf{J}-\mathbf{I}) - \eta \sum_{j=1}^{k}\sum_{i=0}^{\tau-1} \xi^j \mathbf{G}_{\tau(k-j)+d+i}(\mathbf{J}-\mathbf{I}) + \eta \sum_{i=0}^{l} \mathbf{G}_{\tau k+d+i} \right\|_F^2$$

$$\leq 3\eta^2 \mathbb{E}_{K(k)} \left[ \xi^{2k} d \left\| \sum_{i=1}^{d-1} \mathbf{G}_i(\mathbf{J}-\mathbf{I}) \right\|_F^2 + d \left\| \sum_{j=1}^{k}\sum_{i=0}^{\tau-1} \xi^j \mathbf{G}_{\tau(k-j)+d+i}(\mathbf{J}-\mathbf{I}) \right\|_F^2 + \sum_{l=\tau-d}^{\tau-1} \left\| \sum_{i=0}^{l} \mathbf{G}_{\tau k+d+i} \right\|_F^2 \right]$$

$$\leq 3\eta^2 \mathbb{E}_{K(k)} \left[ \xi^{2k} d \left\| \sum_{i=1}^{d-1} \mathbf{G}_i \right\|_F^2 + d \left\| \sum_{j=1}^{k}\sum_{i=0}^{\tau-1} \xi^j \mathbf{G}_{\tau(k-j)+d+i} \right\|_F^2 + \sum_{l=\tau-d}^{\tau-1} \left\| \sum_{i=0}^{l} \mathbf{G}_{\tau k+d+i} \right\|_F^2 \right] \quad (26)$$

$$= 3\eta^2 \sum_{m=1}^{M} \left[ \mathbb{E}_{K(k)} \xi^{2k} d \left\| \sum_{i=1}^{d-1} g(x_i^{(m)}) \right\|^2 + d\mathbb{E}_{K(k)} \left\| \sum_{j=1}^{k}\sum_{i=0}^{\tau-1} \xi^j g(x_{\tau(k-j)+d+i}^{(m)}) \right\|^2 + \mathbb{E}_{K(k)} \sum_{l=\tau-d}^{\tau-1} \left\| \sum_{i=0}^{l} g(x_{\tau k+d+i}^{(m)}) \right\|^2 \right]$$

$$= 3\eta^2 d \left[ \underbrace{\sum_{m=1}^{M} \mathbb{E}_{K(k)} \xi^{2k} \left\| \sum_{i=1}^{d-1} g(x_i^{(m)}) \right\|^2}_{T_1} + \underbrace{\sum_{m=1}^{M} \mathbb{E}_{K(k)} \left\| \sum_{j=1}^{k} \xi^j \sum_{i=0}^{\tau-1} g(x_{\tau(k-j)+d+i}^{(m)}) \right\|^2}_{T_2} + \underbrace{\frac{1}{d}\sum_{m=1}^{M}\sum_{l=\tau-d}^{\tau-1} \mathbb{E}_{K(k)} \left\| \sum_{i=0}^{l} g(x_{\tau k+d+i}^{(m)}) \right\|^2}_{T_3} \right.$$

$$\quad (27)$$

where the equation 26 is due to the operator norm of $\mathbf{J} - \mathbf{I}$ is less than 1.

For $T2$, we have

$$\sum_{m=1}^{M} \mathbb{E}_{K(k)} \left\| \sum_{j=1}^{k} \xi^j \sum_{i=0}^{\tau-1} g(x_{\tau(k-j)+d+i}^{(m)}) \right\|^2$$

$$= \sum_{m=1}^{M} \mathbb{E}_{K(k)} \left\| \sum_{j=1}^{k} \xi^j \sum_{i=0}^{\tau-1} \left[ g(x_{\tau(k-j)+d+i}^{(m)}) - \bigtriangledown F(x_{\tau(k-j)+d+i}^{(m)}) \right] + \sum_{j=1}^{k} \xi^j \sum_{i=0}^{\tau-1} \bigtriangledown F(x_{\tau(k-j)+d+i}^{(m)}) \right\|^2$$

$$\leq 2\underbrace{\sum_{m=1}^{M} \mathbb{E}_{K(k)} \left\| \sum_{j=1}^{k} \xi^j \sum_{i=0}^{\tau-1} \left[ g(x_{\tau(k-j)+d+i}^{(m)}) - \bigtriangledown F(x_{\tau(k-j)+d+i}^{(m)}) \right] \right\|^2}_{T_4} + 2\underbrace{\sum_{m=1}^{M} \mathbb{E}_{K(k)} \left\| \sum_{j=1}^{k} \xi^j \sum_{i=0}^{\tau-1} \bigtriangledown F(x_{\tau(k-j)+d+i}^{(m)}) \right\|^2}_{T_5}$$

For the first term $T_4$, since the stochastic gradients are unbiased, all cross terms are zero. Thus, combining with Assumption of bounded variance, we have

$$T_4 = 2\sum_{m=1}^{M}\sum_{j=1}^{k} \xi^{2j} \sum_{i=0}^{\tau-1} \mathbb{E}_{K(k)} \left\| g(x_{\tau(k-j)+d+i}^{(m)}) - \bigtriangledown F(x_{\tau(k-j)+d+i}^{(m)}) \right\|^2$$

$$\leq 2\sum_{m=1}^{M}\sum_{j=1}^{k} \xi^{2j} \sum_{i=0}^{\tau-1} \sigma^2 \leq \frac{2M\tau\sigma^2\xi^2}{1-\xi^2} \quad (28)$$

where equation 28 according to the summation formula of power

$$\sum_{j=1}^{k} \xi^{2j} \leq \sum_{j=1}^{\infty} \xi^{2j} \leq \frac{\xi^2}{1-\xi^2}$$

For the second term $T_5$, we get

$$T_5 = 2 \sum_{j=1}^{k} \xi^{2j} \sum_{i=0}^{\tau-1} \mathbb{E} \left\| \triangledown F(\mathbf{X}_{\tau(k-j)+d+i}^{(m)}) \right\|_F^2 = 2 \sum_{r=0}^{k-1} \xi^{2(k-r)} \sum_{i=0}^{\tau-1} \mathbb{E} \left\| \triangledown F(\mathbf{X}_{\tau r+d+i}^{(m)}) \right\|_F^2$$

Substituting the bounds of $T_4$ and $T_5$ into $T_2$, we have

$$T_2 \le \frac{2M\tau\sigma^2\xi^2}{1-\xi^2} + 2 \sum_{r=0}^{k-1} \xi^{2(k-r)} \sum_{i=0}^{\tau-1} \mathbb{E} \left\| \triangledown F(\mathbf{X}_{\tau r+d+i}^{(m)}) \right\|_F^2$$

For $T1$, we have

$$\sum_{m=1}^{M} \mathbb{E}_{K(k)} \xi^{2k} \left\| \sum_{i=1}^{d-1} g(x_i^{(m)}) \right\|^2$$

$$= \sum_{m=1}^{M} \mathbb{E}_{K(k)} \xi^{2k} \left\| \sum_{i=1}^{d-1} \left[ g(x_i^{(m)}) - \triangledown F(x_i^{(m)}) \right] + \sum_{i=1}^{d-1} \triangledown F(x_i^{(m)}) \right\|^2$$

$$\le \underbrace{2 \sum_{m=1}^{M} \mathbb{E}_{K(k)} \xi^{2k} \left\| \sum_{i=1}^{d-1} \left[ g(x_i^{(m)}) - \triangledown F(x_i^{(m)}) \right] \right\|^2}_{T_6} + \underbrace{2 \sum_{m=1}^{M} \mathbb{E}_{K(k)} \xi^{2k} \left\| \sum_{i=1}^{d-1} \triangledown F(x_i^{(m)}) \right\|^2}_{T_7}$$

For the first term $T_6$, since the stochastic gradients are unbiased, all cross terms are zero. Thus, combining with Assumption of bounded variance, we have

$$T_6 = 2 \sum_{m=1}^{M} \sum_{i=1}^{d-1} \mathbb{E}_{K(k)} \xi^{2k} \left\| g(x_i^{(m)}) - \triangledown F(x_i^{(m)}) \right\|^2 \le 2\xi^{2k} \sum_{m=1}^{M} \sum_{i=1}^{d-1} \sigma^2 = 2\xi^{2k} Md\sigma^2$$

For the second term $T_7$, directly applying Jensen's inequality, we get

$$T_7 = 2 \sum_{m=1}^{M} \mathbb{E}_{K(k)} \xi^{2k} \left\| \sum_{i=1}^{d-1} \triangledown F(x_i^{(m)}) \right\|^2 \le 2d \sum_{m=1}^{M} \mathbb{E}_{K(k)} \xi^{2k} \sum_{i=1}^{d-1} \left\| \triangledown F(x_i^{(m)}) \right\|^2 = 2d\xi^{2k} \sum_{i=1}^{d-1} \|\triangledown F(\mathbf{X}_i)\|_F^2$$

Substituting the bounds of $T_6$ and $T_7$ into $T_1$, we have

$$T_1 \le 2\xi^{2k} Md\sigma^2 + 2d\xi^{2k} \sum_{i=1}^{d-1} \|\triangledown F(\mathbf{X}_i)\|_F^2$$

For $T_3$, we have

$$T_3 = \frac{1}{d} \sum_{m=1}^{M} \sum_{l=\tau-d}^{\tau-1} \mathbb{E}_{K(k)} \left\| \sum_{i=0}^{l} g(x_{\tau k+d+i}^{(m)}) \right\|^2$$

$$= \frac{1}{d} \sum_{m=1}^{M} \sum_{l=\tau-d}^{\tau-1} \mathbb{E}_{K(k)} \left\| \sum_{i=0}^{l} \left( g(x_{\tau k+d+i}^{(m)}) - \triangledown F(x_{\tau k+d+i}^{(m)}) \right) + \sum_{i=0}^{l} \triangledown F(x_{\tau k+d+i}^{(m)}) \right\|^2$$

$$\le \frac{2}{d} \sum_{m=1}^{M} \sum_{l=\tau-d}^{\tau-1} \mathbb{E}_{K(k)} \left\| \sum_{i=0}^{l} \left( g(x_{\tau k+d+i}^{(m)}) - \triangledown F(x_{\tau k+d+i}^{(m)}) \right) \right\|^2 + \frac{2}{d} \sum_{m=1}^{M} \sum_{l=\tau-d}^{\tau-1} \mathbb{E}_{K(k)} \left\| \sum_{i=0}^{l} \triangledown F(x_{\tau k+d+i}^{(m)}) \right\|^2$$

$$\le \frac{2}{d} \sum_{m=1}^{M} \sum_{l=\tau-d}^{\tau-1} \sum_{i=0}^{l} \mathbb{E}_{K(k)} \left\| g(x_{\tau k+d+i}^{(m)}) - \triangledown F(x_{\tau k+d+i}^{(m)}) \right\|^2 + \frac{2}{d} \sum_{m=1}^{M} \sum_{l=\tau-d}^{\tau-1} \sum_{i=0}^{l} \mathbb{E}_{K(k)} \left\| \triangledown F(x_{\tau k+d+i}^{(m)}) \right\|^2$$

$$\le \frac{2}{d} \sum_{m=1}^{M} \sum_{l=\tau-d}^{\tau-1} \sum_{i=0}^{l} \sigma^2 + \frac{2}{d} \sum_{m=1}^{M} \sum_{l=\tau-d}^{\tau-1} \sum_{i=0}^{l} \mathbb{E}_{K(k)} \left\| \triangledown F(x_{\tau k+d+i}^{(m)}) \right\|^2$$

$$= \frac{2M}{d} \sum_{l=\tau-d}^{\tau-1} \sum_{i=0}^{l} \sigma^2 + \frac{2}{d} \sum_{m=1}^{M} \sum_{l=\tau-d}^{\tau-1} \sum_{i=0}^{l} \mathbb{E}_{K(k)} \left\| \triangledown F(x_{\tau k+d+i}^{(m)}) \right\|^2$$

Substituting the bounds of $T_1$, $T_2$ and $T_3$ into equation 27, we have

$$\sum_{i=\tau-d}^{\tau-1} \mathbb{E}_{K(k)} \left\| \mathbf{X}_{\tau k+d} \mathbf{J} - \mathbf{X}_{\tau k+d+i} \right\|_F^2$$

$$\leq 3\eta^2 d \left[ 2\xi^{2k} M d\sigma^2 + \frac{2M\tau\sigma^2\xi^2}{1-\xi^2} + 2d\xi^{2k} \sum_{i=1}^{d-1} \|\nabla F(\mathbf{X}_i)\|_F^2 + 2\sum_{r=0}^{k-1} \xi^{2(k-r)} \sum_{i=0}^{\tau-1} \mathbb{E} \left\| \nabla F(\mathbf{X}_{\tau r+d+i}^{(m)}) \right\|_F^2 \right.$$

$$\left. + \frac{2M}{d} \sum_{l=\tau-d}^{\tau-1} \sum_{i=0}^{l} \sigma^2 + \frac{2}{d} \sum_{m=1}^{M} \sum_{l=\tau-d}^{\tau-1} \sum_{i=0}^{l} \mathbb{E}_{K(k)} \left\| \nabla F(x_{\tau k+d+i}^{(m)}) \right\|^2 \right]$$

And in the same way, we have

$$\sum_{l=0}^{\tau-1-d} \mathbb{E}_{K(k)} \left\| \mathbf{X}_{\tau k+d} \mathbf{J} - \mathbf{X}_{\tau k+d+l} \right\|_F^2$$

$$\leq 3\eta^2 (\tau-d) \left[ 2\xi^{2k} M d\sigma^2 + \frac{2M\tau\sigma^2\xi^2}{1-\xi^2} + 2d\xi^{2k} \sum_{i=1}^{d-1} \|\nabla F(\mathbf{X}_i)\|_F^2 + 2\sum_{r=0}^{k-1} \xi^{2(k-r)} \sum_{i=0}^{\tau-1} \mathbb{E} \left\| \nabla F(\mathbf{X}_{\tau r+d+i}^{(m)}) \right\|_F^2 \right.$$

$$\left. + \frac{2M}{\tau-d} \sum_{l=0}^{\tau-1-d} \sum_{i=0}^{l} \sigma^2 + \frac{2}{\tau-d} \sum_{m=1}^{M} \sum_{l=0}^{\tau-1-d} \sum_{i=0}^{l} \mathbb{E}_{K(k)} \left\| \nabla F(x_{\tau k+d+i}^{(m)}) \right\|^2 \right]$$

Then, summing over all periods from $k=0$ to $k=K$, where $K$ is the total global iterations:

$$\sum_{k=1}^{K} \sum_{i=\tau-d}^{\tau-1} \mathbb{E}_{K(k)} \left\| \mathbf{X}_{\tau k+d} \mathbf{J} - \mathbf{X}_{\tau k+d+i} \right\|_F^2$$

$$\leq 3\eta^2 d \sum_{k=1}^{K} \left[ 2\xi^{2k} M d\sigma^2 + \frac{2M\tau\sigma^2\xi^2}{1-\xi^2} + 2d\xi^{2k} \sum_{i=1}^{d-1} \|\nabla F(\mathbf{X}_i)\|_F^2 + 2\sum_{r=0}^{k-1} \xi^{2(k-r)} \sum_{i=0}^{\tau-1} \mathbb{E} \left\| \nabla F(\mathbf{X}_{\tau r+d+i}^{(m)}) \right\|_F^2 \right.$$

$$\left. + \frac{2M}{d} \sum_{l=\tau-d}^{\tau-1} \sum_{i=0}^{l} \sigma^2 + \frac{2}{d} \sum_{m=1}^{M} \sum_{l=\tau-d}^{\tau-1} \sum_{i=0}^{l} \mathbb{E}_{K(k)} \left\| \nabla F(x_{\tau k+d+i}^{(m)}) \right\|^2 \right]$$

$$\leq 6\eta^2 d \frac{\xi^2}{1-\xi^2} \left[ 2M d\sigma^2 + 2d \sum_{i=1}^{d-1} \|\nabla F(\mathbf{X}_i)\|_F^2 \right] + \frac{6\eta^2 d\tau\sigma^2\xi^2 MK}{1-\xi^2} + 6\eta^2 MK \sum_{l=\tau-d}^{\tau-1} \sum_{i=0}^{l} \sigma^2$$

$$+ 6\eta^2 d \sum_{k=1}^{K} \sum_{r=0}^{k-1} \xi^{2(k-r)} \sum_{i=0}^{\tau-1} \mathbb{E} \left\| \nabla F(\mathbf{X}_{\tau r+d+i}^{(m)}) \right\|_F^2 + 6\eta^2 \sum_{k=1}^{K} \sum_{l=\tau-d}^{\tau-1} \sum_{i=0}^{l} \mathbb{E}_{K(k)} \left\| \nabla F(\mathbf{X}_{\tau k+d+i}) \right\|^2$$

$$\tag{29}$$

Expanding the summation, we have

$$\sum_{k=1}^{K} \sum_{r=0}^{k-1} \xi^{2(k-r)} \sum_{i=0}^{\tau-1} \mathbb{E} \left\| \nabla F(\mathbf{X}_{\tau r+d+i}^{(m)}) \right\|_F^2$$

$$\leq \sum_{r=1}^{K} \left[ \left( \sum_{i=0}^{\tau-1} \mathbb{E} \|\nabla F(\mathbf{X}_{\tau r+d+i})\|_F^2 \right) \left( \sum_{k=r}^{K} \xi^{2(k-r)} \right) \right]$$

$$\leq \sum_{r=1}^{K} \left[ \left( \sum_{i=0}^{\tau-1} \mathbb{E} \|\nabla F(\mathbf{X}_{\tau r+d+i})\|_F^2 \right) \left( \sum_{k=r}^{+\infty} \xi^{2(k-r)} \right) \right]$$

$$\leq \frac{1}{1-\xi^2} \sum_{k=1}^{K} \sum_{i=0}^{\tau-1} \mathbb{E} \|\nabla F(\mathbf{X}_{\tau k+d+i})\|_F^2 \tag{30}$$

And in the same way, we have

$$\sum_{k=1}^{K}\sum_{l=\tau-d}^{\tau-1}\sum_{i=0}^{l}\mathbb{E}\left\|\triangledown F(\mathbf{X}_{\tau k+d+i})\right\|_F^2 \le d\sum_{k=1}^{K}\sum_{i=0}^{\tau-1}\mathbb{E}\left\|\triangledown F(\mathbf{X}_{\tau k+d+i})\right\|_F^2 \tag{31}$$

Plugging equation 30 and equation 31 into equation 29,

$$\sum_{k=1}^{K}\sum_{i=\tau-d}^{\tau-1}\mathbb{E}_{K(k)}\left\|\mathbf{X}_{\tau k+d}\mathbf{J}-\mathbf{X}_{\tau k+d+i}\right\|_F^2$$

$$\le 6\eta^2 d\frac{\xi^2}{1-\xi^2}\left[2Md\sigma^2+2d\sum_{i=1}^{d-1}\left\|\triangledown F(\mathbf{X}_i)\right\|_F^2\right]+\frac{6\eta^2 d\tau\sigma^2\xi^2 MK}{1-\xi^2}+6\eta^2 MK\sum_{l=\tau-d}^{\tau-1}\sum_{i=0}^{l}\sigma^2$$

$$+\frac{6\eta^2 d}{1-\xi^2}\sum_{k=1}^{K}\sum_{i=0}^{\tau-1}\mathbb{E}\left\|\triangledown F(\mathbf{X}_{\tau k+d+i})\right\|_F^2+6\eta^2 d\sum_{k=1}^{K}\sum_{i=0}^{\tau-1}\mathbb{E}\left\|\triangledown F(\mathbf{X}_{\tau k+d+i})\right\|_F^2$$

And in the same way, we have

$$\sum_{k=1}^{K}\sum_{l=0}^{\tau-1-d}\mathbb{E}_{K(k)}\left\|\mathbf{X}_{\tau k+d}\mathbf{J}-\mathbf{X}_{\tau k+d+l}\right\|_F^2$$

$$\le 6\eta^2(\tau-d)\frac{\xi^2}{1-\xi^2}\left[2Md\sigma^2+2d\sum_{i=1}^{d-1}\left\|\triangledown F(\mathbf{X}_i)\right\|_F^2\right]+\frac{6\eta^2(\tau-d)\tau\sigma^2\xi^2 MK}{1-\xi^2}+6\eta^2 MK\sum_{l=\tau-d}^{\tau-1}\sum_{i=0}^{l}\sigma^2$$

$$+\frac{6\eta^2(\tau-d)}{1-\xi^2}\sum_{k=1}^{K}\sum_{i=0}^{\tau-1}\mathbb{E}\left\|\triangledown F(\mathbf{X}_{\tau k+d+i})\right\|_F^2+6\eta^2(\tau-d)\sum_{k=1}^{K}\sum_{i=0}^{\tau-1-d}\mathbb{E}\left\|\triangledown F(\mathbf{X}_{\tau k+d+i})\right\|_F^2$$

Recall the intermediate result equation 11 in the proof of Lemma 1:

$$\mathbb{E}_{K(k)}\left[\frac{1}{K}\sum_{k=1}^{K}\left\|\triangledown F(\mu_{K(k)})\right\|^2\right]$$

$$\le\frac{2\left[F(\mu_1)-F_{inf}\right]}{\eta K(\xi d+\tau-d)}+\frac{\eta 2L\sigma^2\left[\xi^2 d+\tau-d\right]}{M(\xi d+\tau-d)}+\eta^2\frac{6L^2(1+\xi)}{\xi d+\tau-d}\sum_{l=\tau-d}^{\tau-1}\sum_{i=0}^{l}\sigma^2+\eta^2\frac{6d\xi^2 L^2\tau\sigma^2(1+\xi)}{(\xi d+\tau-d)(1-\xi^2)}$$

$$+\frac{\eta^2}{K}\frac{12d\sigma^2 L^2\xi^2(\tau-d)(1+\xi)}{(\xi d+\tau-d)(1-\xi^2)}+\frac{\eta^2}{KM}\frac{12L^2 d\xi^2(\tau-d)(1+\xi)}{(\xi d+\tau-d)(1-\xi^2)}\sum_{i=1}^{d-1}\left\|\triangledown F(\mathbf{X}_i)\right\|_F^2$$

$$+\frac{2L\eta d\xi^2-\xi+\frac{6\xi L^2\eta^2 d+6L^2\eta^2(\tau-d)}{1-\xi^2}+6\xi L^2\eta^2 d}{KM(\xi d+\tau-d)}\sum_{k=1}^{K}\sum_{i=\tau-d}^{\tau-1}\mathbb{E}_{K(k)}\left\|\triangledown F\left(\mathbf{X}_{\tau k+d+i}\right)\right\|_F^2$$

$$+\frac{2L\eta(\tau-d)-\xi+\frac{6\xi L^2\eta^2 d+6L^2\eta^2(\tau-d)}{1-\xi^2}+6\xi L^2\eta^2 d+6L^2\eta^2(\tau-d)}{KM(\xi d+\tau-d)}\sum_{k=1}^{K}\sum_{i=0}^{\tau-1-d}\mathbb{E}_{K(k)}\left\|\triangledown F\left(\mathbf{X}_{\tau k+d+i}\right)\right\|_F^2$$

When the learning rate satisfies the following two formulas at the same time

$$2L\eta d\xi^2-\xi+\frac{6\xi L^2\eta^2 d+6L^2\eta^2(\tau-d)}{1-\xi^2}+6\xi L^2\eta^2 d\le 0$$

$$2L\eta(\tau-d)-\xi+\frac{6\xi L^2\eta^2 d+6L^2\eta^2(\tau-d)}{1-\xi^2}+6\xi L^2\eta^2 d+6L^2\eta^2(\tau-d)\le 0$$

And

$$\sum_{k=1}^{K}\sum_{i=0}^{\tau-1}\mathbb{E}_{K(k)}\left\|\triangledown F\left(\mathbf{X}_{\tau k+d+i}\right)\right\|_F^2=\mathbb{E}_k\sum_{k=1}^{K}\left\|\triangledown F(\mu_k)\right\|^2$$

Thus, we have

$$\mathbb{E}_{K(k)} \left[ \frac{1}{K} \sum_{k=1}^{K} \left\| \nabla F(\mu_{K(k)}) \right\|^2 \right]$$

$$\leq \frac{2 \left[ F(\mu_1) - F_{inf} \right]}{\eta K (\xi d + \tau - d)} + \frac{\eta 2 L \sigma^2 \left[ \xi^2 d + \tau - d \right]}{M(\xi d + \tau - d)} + \eta^2 \frac{6 L^2 (1 + \xi)}{\xi d + \tau - d} \sum_{l=\tau-d}^{\tau-1} \sum_{i=0}^{l} \sigma^2 + \eta^2 \frac{6 d \xi^2 L^2 \tau \sigma^2 (1 + \xi)}{(\xi d + \tau - d)(1 - \xi^2)}$$

$$+ \frac{\eta^2}{K} \frac{12 d \sigma^2 L^2 \xi^2 (\tau - d)(1 + \xi)}{(\xi d + \tau - d)(1 - \xi^2)} + \frac{\eta^2}{KM} \frac{12 L^2 d \xi^2 (\tau - d)(1 + \xi)}{(\xi d + \tau - d)(1 - \xi^2)} \sum_{i=1}^{d-1} \left\| \nabla F(\mathbf{X}_i) \right\|_F^2$$

---

**Corollary 1.** Under sssumptions, if the learning rate is $\eta = \frac{M+V}{M} \sqrt{\frac{M}{K}}$ the average-squared gradient norm after $K$ iterations is bounded by

$$\mathbb{E}_{K(k)} \left[ \frac{1}{K} \sum_{k=1}^{K} \left\| \nabla F(\mu_{K(k)}) \right\|^2 \right]$$

$$\leq \frac{2 \left[ F(\mu_1) - F_{inf} \right]}{\sqrt{MK}(\xi d + \tau - d)} + \frac{1}{\sqrt{MK}} \frac{2 L \sigma^2 \left[ \xi^2 d + \tau - d \right]}{(\xi d + \tau - d)}$$

$$+ \frac{M^2}{K^3} \left( 1 + \frac{V}{M} \right)^4 \frac{6 L^2 (1 + \xi)}{\xi d + \tau - d} \sum_{l=\tau-d}^{\tau-1} \sum_{i=0}^{l} \sigma^2 + \frac{M^2}{K^3} \left( 1 + \frac{V}{M} \right)^4 \frac{6 d \xi^2 L^2 \tau \sigma^2 (1 + \xi)}{(\xi d + \tau - d)(1 - \xi^2)}$$

$$+ \frac{M^2}{K^3} \left( 1 + \frac{V}{M} \right)^4 \frac{12 d \sigma^2 L^2 \xi^2 (\tau - d)(1 + \xi)}{(\xi d + \tau - d)(1 - \xi^2)} + \frac{M}{K^3} \left( 1 + \frac{V}{M} \right)^4 \frac{12 L^2 d \xi^2 (\tau - d)(1 + \xi)}{(\xi d + \tau - d)(1 - \xi^2)} \sum_{i=1}^{d-1} \left\| \nabla F(\mathbf{X}_i) \right\|_F^2$$

If the total iterations $K$ is sufficiently large, then the average-squared gradient norm will be bounded by

$$\mathbb{E} \left[ \frac{1}{K} \sum_{k=1}^{K} \left\| \nabla F(\mu_k) \right\|^2 \right] \leq \frac{2 \left[ F(\mu_1) - F_{inf} \right] + 2 L \sigma^2 \left[ \xi^2 d + \tau - d \right]}{\sqrt{MK}(\xi d + \tau - d)}$$

.

