# OpenReview forum: "Squeezing SGD Parallelization Performance in Distributed Training Using Delayed Averaging"
_ICLR.cc/2022/Conference — ICLR 2022 Submitted_

### Official Review · Reviewer_SC8P · 2021-11-01

**Correctness:** 4
**Technical Novelty And Significance:** 1
**Empirical Novelty And Significance:** 1
**Recommendation:** 3
**Confidence:** 4

**Main Review:**

1. The critical issue is that neither local sgd nor overlapped [1] communication is novel. What's more, combining them is also not novel. As a matter of fact, the proposed method Figure 2 (c) is nearly the same to Fig 2 in [2].

2. I did not find the improvement regarding the wall clock time by hiding the communication in the experiment.

References

[1] Li, Y., Yu, M., Li, S., Avestimehr, S., Kim, N.S. and Schwing, A., Pipe-SGD: A Decentralized Pipelined SGD Framework for Distributed Deep Net Training. NeurIPS 2018.

[2] Wang, Jianyu, Hao Liang, and Gauri Joshi. "Overlap local-SGD: An algorithmic approach to hide communication delays in distributed SGD." In ICASSP 2020-2020 IEEE International Conference on Acoustics, Speech and Signal Processing (ICASSP), pp. 8871-8875. IEEE, 2020.

**Summary Of The Paper:**

The paper proposes to combine local sgd with overlapped communication. A convergence analysis is given. The experiments validates the model performance.

**Summary Of The Review:**

I believe the proposed method is not novel and [2] has proposed a very similar method.

---

### Official Review · Reviewer_MLkr · 2021-11-02

**Correctness:** 4
**Technical Novelty And Significance:** 1
**Empirical Novelty And Significance:** 2
**Recommendation:** 3
**Confidence:** 3

**Main Review:**

In this paper, the authors present DaSGD, which overlaps the computation and communication of distributed training. The paper is well-written and easy to understand.
In a nutshell, the idea is to combine PipeSGD and local SGD, which totally makes sense. The experiment results show good performance.
However, I have the following concerns in the novelty:
1. The idea of "fully parallelize SGD and forward/backward propagations to hide 100\% of gradient communication" is actually not new. PipeSGD [1] proposed using SGD with 1 step of staleness to overlap communication and computation. However, [1] is not cited or discussed in this paper.
2. The combination of PipeSGD and local SGD is also not new. [2,3] both uses such a combination. Basically, I think [3] proposes the combination of PipeSGD and local SGD, and [2] adds communication compression to [3]. Please correct me if I'm wrong, and indicate the major difference between the ideas of this paper and [3].


References:

[1] Li, Youjie et al. “Pipe-SGD: A Decentralized Pipelined SGD Framework for Distributed Deep Net Training.” NeurIPS 2018.

[2] Delay-Tolerant Local SGD for Efficient Distributed Training. https://openreview.net/forum?id=E_U8Zvx7zrf

[3] Wang, Jianyu et al. “Overlap Local-SGD: An Algorithmic Approach to Hide Communication Delays in Distributed SGD.” ICASSP 2020.


**Summary Of The Paper:**

In this paper, the authors present DaSGD, which overlaps the computation and communication of distributed training.
In a nutshell, the idea is to combine PipeSGD and local SGD, which totally makes sense. The experiment results show good performance.

**Summary Of The Review:**

In this paper, the authors present DaSGD, which overlaps the computation and communication of distributed training. The paper is well-written and easy to understand. However, I have some concerns in the novelty.

---

### Official Review · Reviewer_NVwg · 2021-11-02

**Correctness:** 3
**Technical Novelty And Significance:** 2
**Empirical Novelty And Significance:** 2
**Recommendation:** 3
**Confidence:** 3

**Main Review:**

In this paper, authors consider the Distributed SGD with delayed gradients and bounded delays. This assumption on delays allows algorithm to converge with the same rate as Local SGD algorithm.
First of all, I didn't get the problem we try to solve.
Unfortunately, the algorithm itself is not mentioned in the paper only a single line that corresponds to the update rule. From this rule I conclude that all nodes are interconnected since the second term is dependent on the all neighbors' gradients and points. This questions me if this algorithm is good in practice. Yes, the local update happens much more often than the global update, but what is the reason then to use this step instead of simple averaging?



**Summary Of The Paper:**

In this paper, authors propose the modification of distributed SGD algorithm that combines the ideas of Local SGD and Delayed averaging of updates.

**Summary Of The Review:**

I think that this article is poorly written.

First of all, I want to mention that Authors decided to decrease the font size of all important equations and plots to fit the conference size requirements. This makes this article extremely hard to read after printing.

Second, for me as a colorblind person all the plots are extremely unclear, since the difference between the lines is minimal.
Third, I missed the main problem formulation. Moreover, the algorithm itself is not mentioned (only update without any details about hyperparameters selection).

Second thing to mention is a novelty and a practical interest of the algorithm. I understand what is a difference between this algorithm and the local SGD, but I am hesitating in the following. Since during the full update (not local) we communicate completely (M^2) exchanges what is a reason in such a delayed gradient computation. Usually the communication is much more expensive than the gradient computation so the reason on delaying the gradients is not clear for me.  In work (https://arxiv.org/pdf/1806.09429.pdf) authors use delayed gradients almost the same way as in this paper; however it allows to save the total amount of communications.

About the experimental part I have no comments since I failed to recognize the lines difference.

All in all, I find this article in a very draft stage to be published.

---

### Official Review · Reviewer_1qEJ · 2021-11-02

**Correctness:** 3
**Technical Novelty And Significance:** 2
**Empirical Novelty And Significance:** 2
**Recommendation:** 3
**Confidence:** 5

**Main Review:**

Strengths:
This paper considers a very interesting and relevant problem: reducing communication overhead of distributed deep learning methods. Additionally, the approach of using delayed averaging to hide communication overhead is well motivated.

Weaknesses:
1) Misses a large body of literature on the problem, including:
- the whole set of gossip-based methods that have been proposed for this problem
- the whole set of decentralized asynchronous methods that have been proposed for this problem
- co-distillation based methods hiding communication overhead by synchronizing predictions via knowledge-distillation
2) Additionally, the novelty is highly limited. For example, local steps with delayed averaging has already been used in asynchronous decentralized methods. For instance, see \tau-Overlap Stochastic Gradient Push [a] which combines \tau local steps with asynchronous delayed averaging over arbitrary directed communication graphs (not necessarily all-to-all).
3) The proposed method still requires synchronizing workers during forward/backward passes, which does not tolerating variance in inter-processor update times, which even in the case of homogeneous clusters still presents non-trivial slow-downs in the context of machine-learning problems.
4) Numerical results on ImageNet are not convincing since the benchmarks are non-standard for this literature. Specifically, only using a RN18 in IN1k for 20 epochs with a batch-size of 256, does not provide a good indication on the practical utility of the proposed method relative to related work. Not to mention that a numerical comparison is missed with a large body of related work.
5) Minor point, but the heuristic approximation used to determine the lower bound on the delay hyper-parameter, d, does not necessarily scale linearly with the number of workers. For instance, all-reduce collective communication implementations using distance-halving-vector-doubling or ring-based all-reduce only scales logarithmically with the number of workers, not linearly.

[a] Assran et al., Stochastic Gradient Push for Distributed Deep Learning, ICML, 2019.

[b] Vogels et al., RelaySum for Decentralized Deep Learning on Heterogeneous Data, NeurIPS, 2021.

[c] Koloskova et al., Decentralized Deep Learning with Arbitrary Communication Compression, ICLR 2020.

[d] Lian et al., Can decentralized algorithms outperform centralized algorithms? a case study for decentralized parallel stochastic gradient descent, NeurIPS, 2017.

[e] Lian et al., Asynchronous decentralized parallel stochastic gradient descent, ICML 2018.

[f] Tang et al., Decentralized Training over Decentralized Data, ICML 2018.

[g] Assran et al., Advances in asynchronous parallel and distributed optimization, Proceedings of the IEEE, 2020.

[h] Assran and Rabbat, Asynchronous Gradient Push, IEEE Transactions on Automatic Control, 2021.

[i] Assran et al., Gossip-based Actor-Learner Architectures for Deep Reinforcement Learning, NeurIPS, 2020.

[j] Spiridinoff et al., Robust asynchronous stochastic gradient-push: Asymptotically optimal and network-independent performance for strongly convex functions, JMLR,2020.

[k] Pu et al., Asymptotic network independence in distributed stochastic optimization for machine learning: Examining distributed and centralized stochastic gradient descent, IEEE Signal Processing Magazine, 2020.

[l] Sodhani et al., A Closer Look at Codistillation for Distributed Training, arXiv, 2020.

[m] Anil et al., Large scale distributed neural network training through online distillation, ICLR 2018.

- The "for sufficiently large K" is standard but is quite vague. The lower bound on K is quantified in related works.

**Summary Of The Paper:**

This paper proposed DaSGD, an algorithm for large-scale large-batch training of deep neural networks. The algorithm combines Local SGD with delayed averaging steps to hide the communication overhead. However, workers still synchronize their forward/backward passes in each iteration. A convergence rate of O(1/sqrt(K)) to a stationary is provided for smooth non-convex objectives, with respect to the average parameter set across workers. Numerical experiments provided on CIFAR10 and some brief results on ImageNet-1k with a ResNet18 and a batch-size of 256.

**Summary Of The Review:**

It is my opinion that although the problem is quite interesting and relevant, a large body of literature is missed, and the proposed method is not sufficiently novel. Moreover, the theoretical contributions do not provide improvements over the related literature, and could use further specification in certain parts (e.g., although "for sufficiently large K" is a common qualifier, a reasonable lower bound is usually made specific in the the theorem statement). Additionally, I do not find the numerical experiments to provide convincing evidence that the proposed method is indeed better suited for large-batch distributed deep learning than existing approaches.

---

### Decision · Program_Chairs · 2022-01-20

**Decision:**

Reject

**Comment:**

This paper proposes a variant of stochastic gradient descent that parallelizes the algorithm for distributed training via delayed gradient averaging. While the algorithm (DaSGD) proposed is sensible and seems to work, it also seems to miss a lot of related work. As pointed out by one of the reviewers, the class of asynchronous decentralized methods already seem to cover the space of DaSGD, and it's not clear how DaSGD differs from the existing methods in this space. As a result of this lack of comparison to related work, the reviewers recommended that the paper not be accepted at this time, and this evaluation was not challenged by an author response. I agree with this consensus.